# Maximum Mean Discrepancy Gradient Flow

**Michael Arbel**
Gatsby Computational Neuroscience Unit
University College London
michael.n.arbel@gmail.com

**Anna Korba**
Gatsby Computational Neuroscience Unit
University College London
a.korba@ucl.ac.uk

**Adil Salim**
Visual Computing Center
KAUST
adil.salim@kaust.edu.sa

**Arthur Gretton**
Gatsby Computational Neuroscience Unit
University College London
arthur.gretton@gmail.com

## Abstract

We construct a Wasserstein gradient flow of the maximum mean discrepancy (MMD) and study its convergence properties. The MMD is an integral probability metric defined for a reproducing kernel Hilbert space (RKHS), and serves as a metric on probability measures for a sufficiently rich RKHS. We obtain conditions for convergence of the gradient flow towards a global optimum, that can be related to particle transport when optimizing neural networks. We also propose a way to regularize this MMD flow, based on an injection of noise in the gradient. This algorithmic fix comes with theoretical and empirical evidence. The practical implementation of the flow is straightforward, since both the MMD and its gradient have simple closed-form expressions, which can be easily estimated with samples.

## 1 Introduction

We address the problem of defining a gradient flow on the space of probability distributions endowed with the Wasserstein metric, which transports probability mass from a starting distribtion $\nu$ to a target distribution $\mu$. Our flow is defined on the maximum mean discrepancy (MMD) [19], an integral probability metric [33] which uses the unit ball in a characteristic RKHS [43] as its witness function class. Specifically, we choose the function in the witness class that has the largest difference in expectation under $\nu$ and $\mu$: this difference constitutes the MMD. The idea of descending a gradient flow over the space of distributions can be traced back to the seminal work of [24], who revealed that the Fokker-Planck equation is a gradient flow of the Kullback-Leibler divergence. Its time-discretization leads to the celebrated Langevin Monte Carlo algorithm, which comes with strong convergence guarantees (see [14, 15]), but requires the knowledge of an analytical form of the target $\mu$. A more recent gradient flow approach, Stein Variational Gradient Descent (SVGD) [29], also leverages this analytical $\mu$.

The study of particle flows defined on the MMD relates to two important topics in modern machine learning. The first is in training Implicit Generative Models, notably generative adversarial networks [18]. Integral probability metrics have been used extensively as critic functions in this setting: these include the Wasserstein distance [3, 17, 22] and maximum mean discrepancy [2, 4, 5, 16, 27, 28]. In [32, Section 3.3], a connection between IGMs and particle transport is proposed, where it is shown that gradient flow on the witness function of an integral probability metric takes a similar form to the generator update in a GAN. The critic IPM in this case is the Kernel Sobolev Discrepancy (KSD), which has an additional gradient norm constraint on the witness function compared with the MMD. It is intended as an approximation to the negative Sobolev distance from the optimal transport literature [35, 36, 45]. There remain certain differences between gradient flow and GAN training, however. First, and most obviously, gradient flow can be approximated by representing $\nu$ as a set of particles,

whereas in a GAN $\nu$ is the output of a generator network. The requirement that this generator network be a smooth function of its parameters causes a departure from pure particle flow. Second, in modern implementations [2, 5, 27], the kernel used in computing the critic witness function for an MMD GAN critic is parametrized by a deep network, and an alternating optimization between the critic parameters and the generator parameters is performed. Despite these differences, we anticipate that the theoretical study of MMD flow convergence will provide helpful insights into conditions for GAN convergence, and ultimately, improvements to GAN training algorithms.

Regarding the second topic, we note that the properties of gradient descent for large neural networks have been modeled using the convergence towards a global optimum of particle transport in the population limit, when the number of particles goes to infinity [12, 31, 38, 41]. In particular, [37] show that gradient descent on the parameters of a neural network can also be seen as a particle transport problem, which has as its population limit a gradient flow of a functional defined for probability distributions over the parameters of the network. This functional is in general non-convex, which makes the convergence analysis challenging. The particular structure of the MMD allows us to relate its gradient flow to neural network optimization in a well-specified regression setting similar to [12, 37] (we make this connection explicit in Appendix F).

Our main contribution in this work is to establish conditions for convergence of MMD gradient flow to its *global optimum*. We give detailed descriptions of MMD flow for both its continuous-time and discrete instantiations in Section 2. In particular, the MMD flow may employ a sample approximation for the target $\mu$: unlike e.g. Langevin Monte Carlo or SVGD, it does not require $\mu$ in analytical form. Global convergence is especially challenging to prove: while for functionals that are *displacement convex*, the gradient flow can be shown to converge towards a global optimum [1], the case of non-convex functionals, like the MMD, requires different tools. A modified gradient flow is proposed in [37] that uses particle birth and death to reach global optimality. Global optimality may also be achieved simply by teleporting particles from $\nu$ to $\mu$, as occurs for the Sobolev Discrepancy flow absent a kernel regulariser [32, Theorem 4, Appendix D]. Note, however, that the regularised Kernel Sobolev Discrepancy flow does not rely on teleportation.

Our approach takes inspiration in particular from [7], where it is shown that although the 1-Wasserstein distance is non-convex, it can be optimized up to some barrier that depends on the diameter of the domain of the target distribution. Similarly to [7], we provide in Section 3 a barrier on the gradient flow of the MMD, although the tightness of this barrier in terms of the target diameter remains to be established. We obtain a further condition on the evolution of the flow to ensure global optimality, and give rates of convergence in that case, however the condition is a strong one: it implies that the negative Sobolev distance between the target and the current particles remains bounded at all times.

We thus propose a way to regularize the MMD flow, based on a noise injection (Section 4) in the gradient, with more tractable theoretical conditions for convergence. Encouragingly, the noise injection is shown in practice to ensure convergence in a simple illustrative case where the original MMD flow fails. Finally, while our emphasis has been on establishing conditions for convergence, we note that MMD gradient flow has a simple $O(MN + N^2)$ implementation for $N$ $\nu$-samples and $M$ $\mu$-samples, and requires only evaluating the gradient of the kernel $k$ on the given samples.

## 2 Gradient flow of the MMD in $W_2$

### 2.1 Construction of the gradient flow

In this section we introduce the gradient flow of the Maximum Mean Discrepancy (MMD) and highlight some of its properties. We start by briefly reviewing the MMD introduced in [19]. We define $\mathcal{X} \subset \mathbb{R}^d$ as the closure of a convex open set, and $\mathcal{P}_2(\mathcal{X})$ as the set of probability distributions on $\mathcal{X}$ with finite second moment, equipped with the 2-Wasserstein metric denoted $W_2$. For any $\nu \in \mathcal{P}_2(\mathcal{X})$, $L_2(\nu)$ is the set of square integrable functions w.r.t. $\nu$. The reader may find a relevant mathematical background in Appendix A.

**Maximum Mean Discrepancy.** Given a characteristic kernel $k : \mathcal{X} \times \mathcal{X} \to \mathbb{R}$, we denote by $\mathcal{H}$ its corresponding RKHS (see [42]). The space $\mathcal{H}$ is a Hilbert space with inner product $\langle ., . \rangle_{\mathcal{H}}$ and norm $\|.\|_{\mathcal{H}}$. We will rely on specific assumptions on the kernel which are given in Appendix B. In particular, Assumption (A) states that the gradient of the kernel, $\nabla k$, is Lipschitz with constant $L$. For such kernels, it is possible to define the Maximum Mean Discrepancy as a distance on $\mathcal{P}_2(\mathcal{X})$.

The MMD can be written as the RKHS norm of the unnormalised *witness function* $f_{\mu,\nu}$ between $\mu$ and $\nu$, which is the difference between the mean embeddings of $\nu$ and $\mu$,

$$MMD(\mu,\nu) = \|f_{\mu,\nu}\|_{\mathcal{H}}, \qquad f_{\nu,\mu}(z) = \int k(x,z)\,\mathrm{d}\nu(x) - \int k(x,z)\,\mathrm{d}\mu(x) \quad \forall z \in \mathcal{X} \quad (1)$$

Throughout the paper, $\mu$ will be fixed and $\nu$ can vary, hence we will only consider the dependence in $\nu$ and denote by $\mathcal{F}(\nu) = \frac{1}{2}MMD^2(\mu,\nu)$. A direct computation [32, Appendix B] shows that for any finite measure $\chi$ such that $\nu + \epsilon\chi \in \mathcal{P}_2(\mathcal{X})$, we have

$$\lim_{\epsilon \to 0} \epsilon^{-1}(\mathcal{F}(\nu + \epsilon\chi) - \mathcal{F}(\nu)) = \int f_{\mu,\nu}(x)d\chi(x). \quad (2)$$

This means that $f_{\mu,\nu}$ is the differential of $\mathcal{F}(\nu)$. Interestingly, $\mathcal{F}(\nu)$ admits a *free-energy* expression:

$$\mathcal{F}(\nu) = \int V(x)\,\mathrm{d}\nu(x) + \frac{1}{2}\int W(x,y)\,\mathrm{d}\nu(x)\,\mathrm{d}\nu(y) + C. \quad (3)$$

where $V$ is a confinement potential, $W$ an interaction potential and $C$ a constant defined by:

$$V(x) = -\int k(x,x')\,\mathrm{d}\mu(x'), \quad W(x,x') = k(x,x'), \quad C = \frac{1}{2}\int k(x,x')\,\mathrm{d}\mu(x)\,\mathrm{d}\mu(x') \quad (4)$$

Formulation (3) and the simple expression of the differential in (2) will be key to construct a gradient flow of $\mathcal{F}(\nu)$, to transport particles. In (4), $V$ reflects the potential generated by $\mu$ and acting on each particle, while $W$ reflects the potential arising from the interactions between those particles.

**Gradient flow of the MMD.** We consider now the problem of transporting mass from an initial distribution $\nu_0$ to a target distribution $\mu$, by finding a continuous path $\nu_t$ starting from $\nu_0$ that converges to $\mu$ while decreasing $\mathcal{F}(\nu_t)$. Such a path should be physically plausible, in that teleportation phenomena are not allowed. For instance, the path $\nu_t = (1 - e^{-t})\mu + e^{-t}\nu_0$ would constantly teleport mass between $\mu$ and $\nu_0$ although it decreases $\mathcal{F}$ since $\mathcal{F}(\nu_t) = e^{-2t}\mathcal{F}(\nu_0)$ [32, Section 3.1, Case 1]. The physicality of the path is understood in terms of classical statistical physics: given an initial configuration $\nu_0$ of $N$ particles, these can move towards a new configuration $\mu$ through successive small transformations, without jumping from one location to another.

Optimal transport theory provides a way to construct such a continuous path by means of the *continuity equation*. Given a vector field $V_t$ on $\mathcal{X}$ and an initial condition $\nu_0$, the continuity equation is a partial differential equation which defines a path $\nu_t$ evolving under the action of the vector field $V_t$, and reads $\partial_t\nu_t = -div(\nu_t V_t)$ for all $t \geq 0$. The reader can find more detailed discussions in Appendix A.2 or [39]. Following [1], a natural choice is to choose $V_t$ as the negative gradient of the differential of $\mathcal{F}(\nu_t)$ at $\nu_t$, since it corresponds to a gradient flow of $\mathcal{F}$ associated with the $W_2$ metric (see Appendix A.3). By (2), we know that the differential of $\mathcal{F}(\nu_t)$ at $\nu_t$ is given by $f_{\mu,\nu_t}$, hence $V_t(x) = -\nabla f_{\mu,\nu_t}(x)$.[1] The gradient flow of $\mathcal{F}$ is then defined by the solution $(\nu_t)_{t\geq 0}$ of

$$\partial_t\nu_t = div(\nu_t\nabla f_{\mu,\nu_t}). \quad (5)$$

Equation (5) is non-linear in that the vector field depends itself on $\nu_t$. This type of equation is associated in the probability theory literature to the so-called McKean-Vlasov process [26, 30],

$$dX_t = -\nabla f_{\mu,\nu_t}(X_t)dt \qquad X_0 \sim \nu_0. \quad (6)$$

In fact, (6) defines a process $(X_t)_{t\geq 0}$ whose distribution $(\nu_t)_{t\geq 0}$ satisfies (5), as shown in Proposition 1. $(X_t)_{t\geq 0}$ can be interpreted as the trajectory of a single particle, starting from an initial random position $X_0$ drawn from $\nu_0$. The trajectory is driven by the velocity field $-\nabla f_{\mu,\nu_t}$, and is affected by other particles. These interactions are captured by the velocity field through the dependence on the current distribution $\nu_t$ of all particles. Existence and uniqueness of a solution to (5) and (6) are guaranteed in the next proposition, whose proof is given Appendix C.1.

**Proposition 1.** *Let $\nu_0 \in \mathcal{P}_2(\mathcal{X})$. Then, under Assumption (A), there exists a unique process $(X_t)_{t\geq 0}$ satisfying the McKean-Vlasov equation in (6) such that $X_0 \sim \nu_0$. Moreover, the distribution $\nu_t$ of $\bar{X}_t$ is the unique solution of (5) starting from $\nu_0$, and defines a gradient flow of $\mathcal{F}$.*

Besides existence and uniqueness of the gradient flow of $\mathcal{F}$, one expects $\mathcal{F}$ to decrease along the path $\nu_t$ and ideally to converge towards 0. The first property, stated in the next proposition, is rather easy to get and is the object of Proposition 2, similar to the result for KSD flow in [32, Section 3.1].

**Proposition 2.** *Under Assumption (A), $\mathcal{F}(\nu_t)$ is decreasing in time and satisfies:*

$$\frac{d\mathcal{F}(\nu_t)}{dt} = -\int \|\nabla f_{\mu,\nu_t}(x)\|^2 \, d\nu_t(x). \tag{7}$$

This property results from (5) and the energy identity in [1, Theorem 11.3.2] and is proved in Appendix C.1. From (7), $\mathcal{F}$ can be seen as a Lyapunov functional for the dynamics defined by (5), since it is decreasing in time. Hence, the continuous-time gradient flow introduced in (5) allows to formally consider the notion of gradient descent on $\mathcal{P}_2(\mathcal{X})$ with $\mathcal{F}$ as a cost function. A time-discretized version of the flow naturally follows, and is provided in the next section.

## 2.2 Euler scheme

We consider here a forward-Euler scheme of (5). For any $T : \mathcal{X} \to \mathcal{X}$ a measurable map, and $\nu \in \mathcal{P}_2(\mathcal{X})$, we denote the pushforward measure by $T_{\#}\nu$ (see Appendix A.2). Starting from $\nu_0 \in \mathcal{P}_2(\mathcal{X})$ and using a step-size $\gamma > 0$, a sequence $\nu_n \in \mathcal{P}_2(\mathcal{X})$ is given by iteratively applying

$$\nu_{n+1} = (I - \gamma \nabla f_{\mu,\nu_n})_{\#}\nu_n. \tag{8}$$

For all $n \geq 0$, equation (8) is the distribution of the process defined by

$$X_{n+1} = X_n - \gamma \nabla f_{\mu,\nu_n}(X_n) \qquad X_0 \sim \nu_0. \tag{9}$$

The asymptotic behavior of (8) as $n \to \infty$ will be the object of Section 3. For now, we provide a guarantee that the sequence $(\nu_n)_{n \in \mathbb{N}}$ approaches $(\nu_t)_{t \geq 0}$ as the step-size $\gamma \to 0$.

**Proposition 3.** *Let $n \geq 0$. Consider $\nu_n$ defined in (8), and the interpolation path $\rho_t^\gamma$ defined as: $\rho_t^\gamma = (I - (t - n\gamma)\nabla f_{\mu,\nu_n})_{\#}\nu_n, \forall t \in [n\gamma, (n+1)\gamma)$. Then, under Assumption (A), $\forall\, T > 0$,*

$$W_2(\rho_t^\gamma, \nu_t) \leq \gamma C(T) \quad \forall t \in [0, T] \tag{10}$$

*where $C(T)$ is a constant that depends only on $T$.*

A proof of Proposition 3 is provided in Appendix C.2 and relies on standard techniques to control the discretization error of a forward-Euler scheme. Proposition 3 means that $\nu_n$ can be linearly interpolated giving rise to a path $\rho_t^\gamma$ which gets arbitrarily close to $\nu_t$ on bounded intervals. Note that as $T \to \infty$ the bound $C(T)$ it is expected to blow up. However, this result is enough to show that (8) is indeed a discrete-time flow of $\mathcal{F}$. In fact, provided that $\gamma$ is small enough, $\mathcal{F}(\nu_n)$ is a decreasing sequence, as shown in Proposition 4.

**Proposition 4.** *Under Assumption (A), and for $\gamma \leq 2/3L$, the sequence $\mathcal{F}(\nu_n)$ is decreasing, and*

$$\mathcal{F}(\nu_{n+1}) - \mathcal{F}(\nu_n) \leq -\gamma(1 - \frac{3\gamma}{2}L)\int \|\nabla f_{\mu,\nu_n}(x)\|^2 \, d\nu_n(x), \quad \forall n \geq 0.$$

Proposition 4, whose proof is given in Appendix C.2, is a discrete analog of Proposition 2. In fact, (8) is intractable in general as it requires the knowledge of $\nabla f_{\mu,\nu_n}$ (and thus of $\nu_n$) exactly at each iteration $n$. Nevertheless, we present in Section 4.2 a practical algorithm using a finite number of samples which is provably convergent towards (8) as the sample-size increases. We thus begin by studying the convergence properties of the time discretized MMD flow (8) in the next section.

## 3 Convergence properties of the MMD flow

We are interested in analyzing the asymptotic properties of the gradient flow of $\mathcal{F}$. Although we know from Propositions 2 and 4 that $\mathcal{F}$ decreases in time, it can very well converge to local minima. One way to see this is by looking at the equilibrium condition for (7). As a non-negative and decreasing function, $t \mapsto \mathcal{F}(\nu_t)$ is guaranteed to converge towards a finite limit $l \geq 0$, which implies in turn that the r.h.s. of (7) converges to 0. If $\nu_t$ happens to converge towards some distribution $\nu^*$, it is possible to show that the equilibrium condition (11) must hold [31, Prop. 2] ,

$$\int \|\nabla f_{\mu,\nu^*}(x)\|^2 \, d\nu^*(x) = 0. \tag{11}$$

Condition (11) does not necessarily imply that $\nu^*$ is a global optimum unless when the loss function has a particular structure [11]. For instance, this would hold if the kernel is linear in at least one of its dimensions. However, when a characteristic kernel is required (to ensure the MMD is a distance), such a structure can't be exploited. Similarly, the claim that KSD flow converges globally, [32, Prop. 3, Appendix B.1], requires an assumption [32, Assump. A] that excludes local minima which are not global (see Appendix D.1; recall KSD is related to MMD). Global convergence of the flow is harder to obtain, and will be the topic of this section. The main challenge is the lack of convexity of $\mathcal{F}$ w.r.t. the Wassertein metric. We show that $\mathcal{F}$ is merely $\Lambda$-convex, and that standard optimization techniques only provide a loose bound on its asymptotic value. We next exploit a Lojasiewicz type inequality to prove convergence to the global optimum provided that a particular quantity remains bounded at all times.

## 3.1 Optimization in a $(W_2)$ non-convex setting

The *displacement convexity* of a functional $\mathcal{F}$ is an important criterion in characterizing the convergence of its Wasserstein gradient flow. Displacement convexity states that $t \mapsto \mathcal{F}(\rho_t)$ is a convex function whenever $(\rho_t)_{t \in [0,1]}$ is a path of minimal length between two distributions $\mu$ and $\nu$ (see Definition 2). Displacement convexity should not be confused with *mixture convexity*, which corresponds to the usual notion of convexity. As a matter of fact, $\mathcal{F}$ is mixture convex in that it satisfies: $\mathcal{F}(t\nu + (1-t)\nu') \leq t\mathcal{F}(\nu) + (1-t)\mathcal{F}(\nu')$ for all $t \in [0,1]$ and $\nu, \nu' \in \mathcal{P}_2(\mathcal{X})$ (see Lemma 25). Unfortunately, $\mathcal{F}$ *is not displacement convex*. Instead, $\mathcal{F}$ only satisfies a weaker notion of displacement convexity called $\Lambda$-displacement convexity, given in Definition 4 (Appendix A.4).

**Proposition 5.** *Under Assumptions (A) to (C), $\mathcal{F}$ is $\Lambda$-displacement convex, and satisfies*

$$\mathcal{F}(\rho_t) \leq (1-t)\mathcal{F}(\nu) + t\mathcal{F}(\nu') - \int_0^1 \Lambda(\rho_s, v_s) G(s,t) \, \mathrm{d}s \tag{12}$$

*for all $\nu, \nu' \in \mathcal{P}_2(\mathcal{X})$ and any displacement geodesic $(\rho_t)_{t \in [0,1]}$ from $\nu$ to $\nu'$ with velocity vectors $(v_t)_{t \in [0,1]}$. The functional $\Lambda$ is defined for any pair $(\rho, v)$ with $\rho \in \mathcal{P}_2(\mathcal{X})$ and $\|v\| \in L_2(\rho)$,*

$$\Lambda(\rho, v) = \left\| \int v(x).\nabla_x k(x,.) \, \mathrm{d}\rho(x) \right\|_{\mathcal{H}}^2 - \sqrt{2}\lambda d\mathcal{F}(\rho)^{\frac{1}{2}} \int \|v(x)\|^2 \, \mathrm{d}\rho(x), \tag{13}$$

*where $(s,t) \mapsto G(s,t) = s(1-t)\mathbb{1}\{s \leq t\} + t(1-s)\mathbb{1}\{s \geq t\}$ and $\lambda$ is defined in Assumption (C).*

Proposition 5 can be obtained by computing the second time derivative of $\mathcal{F}(\rho_t)$, which is then lower-bounded by $\Lambda(\rho_t, v_t)$ (see Appendix D.2). In (13), the map $\Lambda$ is a difference of two non-negative terms: thus $\int_0^1 \Lambda(\rho_s, v_s) G(s,t) \, \mathrm{d}s$ can become negative, and displacement convexity does not hold in general. [8, Theorem 6.1] provides a convergence when only $\Lambda$-displacement convexity holds as long as either the potential or the interaction term is convex enough. In fact, as mentioned in [8, Remark 6.4], the convexity of either term could compensate for a lack of convexity of the other. Unfortunately, this cannot be applied for MMD since both terms involve the same kernel but with opposite signs. Hence, even under convexity of the kernel, a concave term appears and cancels the effect of the convex term. Moreover, the requirement that the kernel be positive semi-definite makes it hard to construct interesting convex kernels. However, it is still possible to provide an upper bound on the asymptotic value of $\mathcal{F}(\nu_n)$ when $(\nu_n)_{n \in \mathbb{N}}$ are obtained using (8). This bound is given in Theorem 6, and depends on a scalar $K(\rho^n) := \int_0^1 \Lambda(\rho_s^n, v_s^n)(1-s) \, \mathrm{d}s$, where $(\rho_s^n)_{s \in [0,1]}$ is a *constant speed displacement geodesic* from $\nu_n$ to the optimal value $\mu$, with velocity vectors $(v_s^n)_{s \in [0,1]}$ of constant norm.

**Theorem 6.** *Let $\bar{K}$ be the average of $(K(\rho^j))_{0 \leq j \leq n}$. Under Assumptions (A) to (C) and if $\gamma \leq 1/3L$,*

$$\mathcal{F}(\nu_n) \leq \frac{W_2^2(\nu_0, \mu)}{2\gamma n} - \bar{K}. \tag{14}$$

Theorem 6 is obtained using techniques from optimal transport and optimization. It relies on Proposition 5 and Proposition 4 to prove an *extended variational inequality* (see Proposition 16), and concludes using a suitable Lyapunov function. A full proof is given in Appendix D.3. When $\bar{K}$ is non-negative, one recovers the usual convergence rate as $O(\frac{1}{n})$ for the gradient descent algorithm. However, $\bar{K}$ can be negative in general, and would therefore act as a barrier on the optimal value

that $\mathcal{F}(\nu_n)$ can achieve when $n \to \infty$. In that sense, the above result is similar to [7, Theorem 6.9]. Theorem 6 only provides a loose bound, however. In Section 3.2 we show global convergence, under the boundedness at all times $t$ of a specific distance between $\nu_t$ and $\mu$.

## 3.2 A condition for global convergence

The lack of convexity of $\mathcal{F}$, as shown in Section 3.1, suggests that a finer analysis of the convergence should be performed. One strategy is to provide estimates for the dynamics in Proposition 2 using differential inequalities which can be solved using the Gronwall's lemma (see [34]). Such inequalities are known in the optimization literature as Lojasiewicz inequalities (see [6]), and upper-bound $\mathcal{F}(\nu_t)$ by the absolute value of its time derivative $\int \|\nabla f_{\mu,\nu_t}(x)\|^2 \, \mathrm{d}\nu_t(x)$. The latter is the squared *weighted Sobolev semi-norm* of $f_{\mu,\nu_t}$ (see Appendix D.4), also written $\|f_{\mu,\nu_t}\|_{\dot{H}(\nu_t)}$. Thus one needs to find a relationship between $\mathcal{F}(\nu_t) = \frac{1}{2}\|f_{\mu,\nu_t}\|_{\mathcal{H}}^2$ and $\|f_{\mu,\nu_t}\|_{\dot{H}(\nu_t)}$. For this purpose, we consider the *weighted negative Sobolev distance* on $\mathcal{P}_2(\mathcal{X})$, defined by duality using $\|.\|_{\dot{H}(\nu)}$ (see also [36]).

**Definition 1.** *Let $\nu \in \mathcal{P}_2(\mathbf{x})$, with its corresponding weighted Sobolev semi-norm $\|.\|_{\dot{H}(\nu)}$. The weighted negative Sobolev distance $\|p-q\|_{\dot{H}^{-1}(\nu)}$ between any $p$ and $q$ in $\mathcal{P}_2(\mathbf{x})$ is defined as*

$$\|p-q\|_{\dot{H}^{-1}(\nu)} = \sup_{f \in L_2(\nu), \|f\|_{\dot{H}(\nu)} \leq 1} \left| \int f(x) \, \mathrm{d}p(x) - \int f(x) \, \mathrm{d}q(x) \right| \tag{15}$$

*with possibly infinite values.*

Equation (59) plays a fundamental role in dynamic optimal transport. It can be seen as the minimum kinetic energy needed to advect the mass $\nu$ to $q$ (see [32]). It is shown in Appendix D.4 that

$$\|f_{\mu,\nu_t}\|_{\mathcal{H}}^2 \leq \|f_{\mu,\nu_t}\|_{\dot{H}(\nu_t)} \|\mu - \nu_t\|_{\dot{H}^{-1}(\nu_t)}. \tag{16}$$

Provided that $\|\mu - \nu_t\|_{\dot{H}^{-1}(\nu_t)}$ remains bounded by some positive constant $C$ at all times, (16) leads to a functional version of Lojasiewicz inequality for $\mathcal{F}$. It is then possible to use the general strategy explained earlier to prove the convergence of the flow to a global optimum:

**Proposition 7.** *Under Assumption (A),*

*(i) If $\|\mu - \nu_t\|_{\dot{H}^{-1}(\nu_t)}^2 \leq C$, for all $t \geq 0$, then: $\mathcal{F}(\nu_t) \leq \frac{C}{C\mathcal{F}(\nu_0)^{-1} + 4t}$,*

*(ii) If $\|\mu - \nu_n\|_{\dot{H}^{-1}(\nu_n)}^2 \leq C$ for all $n \geq 0$, then: $\mathcal{F}(\nu_n) \leq \frac{C}{C\mathcal{F}(\nu_0)^{-1} + 4\gamma(1 - \frac{3}{2}\gamma L)n}$.*

Proofs of Proposition 7 (i) and (ii) are direct consequences of Propositions 2 and 4 and the bounded energy assumption: see Appendix D.4. The fact that (59) appears in the context of Wasserstein flows of $\mathcal{F}$ is not a coincidence. Indeed, (59) is a linearization of the Wasserstein distance (see [35, 36] and Appendix D.6). Gradient flows of $\mathcal{F}$ defined under different metrics would involve other kinds of distances instead of (59). For instance, [37] consider gradient flows under a hybrid metric (a mixture between the Wasserstein distance and KL divergence), where convergence rates can then be obtained provided that the chi-square divergence $\chi^2(\mu\|\nu_t)$ remains bounded. As shown in Appendix D.6, $\chi^2(\mu\|\nu_t)^{\frac{1}{2}}$ turns out to linearize $KL(\mu\|\nu_t)^{\frac{1}{2}}$ when $\mu$ and $\nu_t$ are close. Hence, we conjecture that gradient flows of $\mathcal{F}$ under a metric $d$ can be shown to converge when the linearization of the metric remains bounded. This can be verified on simple examples for $\|\mu - \nu_t\|_{\dot{H}^{-1}(\nu_t)}$ as discussed in Appendix D.5. However, it remains hard to guarantee this condition in general. One possible approach could be to regularize $\mathcal{F}$ using an estimate of (59). Indeed, [32] considers the gradient flow of a regularized version of the negative Sobolev distance which can be written in closed form, and shows that this decreases the MMD. Combing both losses could improve the overall convergence properties of the MMD, albeit at additional computational cost. In the next section, we propose a different approach to improve the convergence, and a particle-based algorithm to approximate the MMD flow in practice.

# 4 A practical algorithm to descend the MMD flow

## 4.1 A noisy update as a regularization

We showed in Section 3.1 that $\mathcal{F}$ is a non-convex functional, and derived a condition in Section 3.2 to reach the global optimum. We now address the case where such a condition does not necessarily hold,

and provide a regularization of the gradient flow to help achieve global optimality in this scenario. Our starting point will be the equilibrium condition in (11). If an equilibrium $\nu^*$ that satisfies (11) happens to have a positive density, then $f_{\mu,\nu^*}$ would be constant everywhere. This in turn would mean that $f_{\mu,\nu^*} = 0$ when the RKHS does not contain constant functions, as for a gaussian kernel [44, Corollary 4.44]. Hence, $\nu^*$ would be a global optimum since $\mathcal{F}(\nu^*) = 0$. The limit distribution $\nu^*$ might be singular, however, and can even be a dirac distribution [31, Theorem 6]. Although the gradient $\nabla f_{\mu,\nu^*}$ is not identically 0 in that case, (11) only evaluates it on the support $\nu^*$, on which $\nabla f_{\mu,\nu^*} = 0$ holds. Hence a possible fix would be to make sure that the unnormalised witness gradient is also evaluated at points outside of the support of $\nu^*$. Here, we propose to regularize the flow by injecting noise into the gradient during updates of (9),

$$X_{n+1} = X_n - \gamma \nabla f_{\mu,\nu_n}(X_n + \beta_n U_n), \qquad n \geq 0, \tag{17}$$

where $U_n$ is a standard gaussian variable and $\beta_n$ is the noise level at $n$. Compared to (8), the sample here is first blurred before evaluating the gradient. Intuitively, if $\nu_n$ approaches a local optimum $\nu^*$, $\nabla f_{\mu,\nu_n}$ would be small on the support of $\nu_n$ but it might be much larger outside of it, hence evaluating $\nabla f_{\mu,\nu_n}$ outside the support of $\nu_n$ can help in escaping the local minimum. The stochastic process (17) is different from adding a diffusion term to (5). The latter case would correspond to regularizing $\mathcal{F}$ using an entropic term as in [31, 40] (see also Appendix A.5 on the Langevin diffusion) and was shown to converge to a global optimum that is in general different from the global minmum of the un-regularized loss. Eq. (17) is also different from [9, 13], where $\mathcal{F}$ (and thus its associated velocity field) is regularized by convolving the interaction potential $W$ in (4) with a mollifier. The optimal solution of a regularized version of the functional $\mathcal{F}$ will be generally different from the non-regularized one, however, which is not desirable in our setting. Eq. (17) is more closely related to the *continuation methods* [10, 20, 21] and *graduated optimization* [23] used for non-convex optimization in Euclidian spaces, which inject noise into the gradient of a loss function $F$ at each iteration. The key difference is the dependence of $f_{\mu,\nu_n}$ of $\nu_n$, which is inherently due to functional optimization. We show in Proposition 8 that (17) attains the global minimum of $\mathcal{F}$ provided that the level of the noise is well controlled, with the proof given in Appendix E.1.

**Proposition 8.** *Let $(\nu_n)_{n \in \mathbb{N}}$ be defined by (17) with an initial $\nu_0$. Denote $\mathcal{D}_{\beta_n}(\nu_n) = \mathbb{E}_{x \sim \nu_n, u \sim g}[\|\nabla f_{\mu,\nu_n}(x + \beta_n u)\|^2]$ with $g$ the density of the standard gaussian distribution. Under Assumptions (A) and (D), and for a choice of $\beta_n$ such that*

$$8\lambda^2 \beta_n^2 \mathcal{F}(\nu_n) \leq \mathcal{D}_{\beta_n}(\nu_n), \tag{18}$$

*the following inequality holds:* $\qquad \mathcal{F}(\nu_{n+1}) - \mathcal{F}(\nu_n) \leq -\frac{\gamma}{2}(1 - 3\gamma L)\mathcal{D}_{\beta_n}(\nu_n),$ $\qquad$ (19)

*where $\lambda$ and $L$ are defined in Assumptions (A) and (D) and depend only on the choice of the kernel. Moreover if $\sum_{i=0}^{n} \beta_i^2 \to \infty$, then*

$$\mathcal{F}(\nu_n) \leq \mathcal{F}(\nu_0)e^{-4\lambda^2 \gamma(1-3\gamma L)\sum_{i=0}^{n} \beta_i^2}. \tag{20}$$

A particular case where $\sum_{i=0}^{n} \beta_i^2 \to \infty$ holds is when $\beta_n$ decays as $1/\sqrt{n}$ while still satisfying (18). In this case, convergence occurs in polynomial time. At each iteration, the level of the noise needs to be adjusted such that the gradient is not too blurred. This ensures that each step decreases the loss functional. However, $\beta_n$ does not need to decrease at each iteration: it could increase adaptively whenever needed. For instance, when the sequence gets closer to a local optimum, it is helpful to increase the level of the noise to probe the gradient in regions where its value is not flat. Note that for $\beta_n = 0$ in (19), we recover a similar bound to Proposition 4.

## 4.2 The sample-based approximate scheme

We now provide a practical algorithm to implement the noisy updates in the previous section, which employs a discretization in space. The update (17) involves computing expectations of the gradient of the kernel $k$ w.r.t the target distribution $\mu$ and the current distribution $\nu_n$ at each iteration $n$. This suggests a simple approximate scheme, based on samples from these two distributions, where at each iteration $n$, we model a system of $N$ interacting particles $(X_n^i)_{1 \leq i \leq N}$ and their empirical distribution in order to approximate $\nu_n$. More precisely, given i.i.d. samples $(X_0^i)_{1 \leq i \leq N}$ and $(Y^m)_{1 \leq m \leq M}$ from $\nu_0$ and $\mu$ and a step-size $\gamma$, the approximate scheme iteratively updates the $i$-th particle as

$$X_{n+1}^i = X_n^i - \gamma \nabla f_{\hat{\mu},\hat{\nu}_n}(X_n^i + \beta_n U_n^i), \tag{21}$$

where $U_n^i$ are i.i.d standard gaussians and $\hat{\mu}$, $\hat{\nu}_n$ denote the empirical distributions of $(Y^m)_{1 \leq m \leq M}$ and $(X_n^i)_{1 \leq i \leq N}$, respectively. It is worth noting that for $\beta_n = 0$, (21) is equivalent to gradient descent over the particles $(X_n^i)$ using a sample based version of the MMD. Implementing (21) is straightforward as it only requires to evaluate the gradient of $k$ on the current particles and target samples. Pseudocode is provided in Algorithm 1. The overall computational cost of the algorithm at each iteration is $O((M + N)N)$ with $O(M + N)$ memory. The computational cost becomes $O(M + N)$ when the kernel is approximated using random features, as is the case for regression with neural networks (Appendix F). This is in contrast to the cubic cost of the flow of the KSD [32], which requires solving a linear system at each iteration. The cost can also be compared to the algorithm in [40], which involves computing empirical CDF and quantile functions of random projections of the particles.

The approximation scheme in (21) is a particle version of (17), so one would expect it to converge towards its population version (17) as $M$ and $N$ goes to infinity. This is shown below.

**Theorem 9.** *Let $n \geq 0$ and $T > 0$. Let $\nu_n$ and $\hat{\nu}_n$ defined by (8) and (21) respectively. Suppose Assumption (A) holds and that $\beta_n < B$ for all $n$, for some $B > 0$. Then for any $\frac{T}{\gamma} \geq n$:*

$$\mathbb{E}\left[W_2(\hat{\nu}_n, \nu_n)\right] \leq \frac{1}{4} \left( \frac{1}{\sqrt{N}}(B + var(\nu_0)^{\frac{1}{2}})e^{2LT} + \frac{1}{\sqrt{M}}var(\mu)^{\frac{1}{2}}) \right) (e^{4LT} - 1)$$

Theorem 9 controls the propagation of the chaos at each iteration, and uses techniques from [25]. Notice also that these rates remain true when no noise is added to the updates, i.e. for the original flow when $B = 0$. A proof is provided in Appendix E.2. The dependence in $\sqrt{M}$ underlines the fact that our procedure could be interesting as a sampling algorithm when one only has access to $M$ samples of $\mu$ (see Appendix A.5 for a more detailed discussion).

**Experiments**

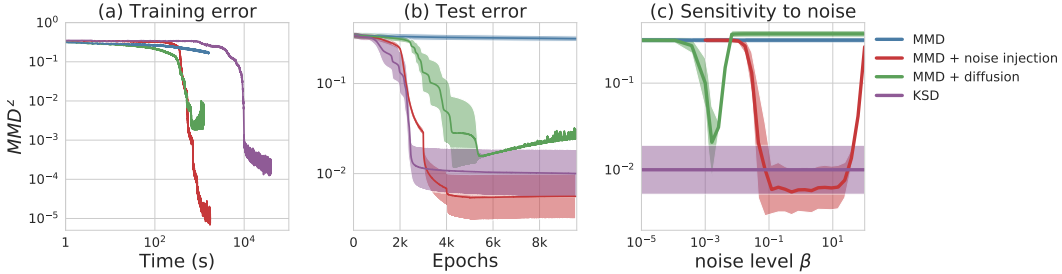

Figure 1: Comparison between different training methods for student-teacher ReLU networks with gaussian output non-linearity and synthetic data uniform on a hyper-sphere. In blue, (21) is used without noise $\beta_n = 0$ while in red noise is added with the following schedule: $\beta_0 > 0$ and $\beta_n$ is decreased by half after every $10^3$ epochs. In green, a diffusion term is added to the particles with noise level kept fixed during training ($\beta_n = \beta_0$). In purple, the KSD is used as a cost function instead of the MMD. In all cases, the kernel is estimated using random features (RF) with a batch size of $10^2$. Best step-size was selected for each method from $\{10^{-3}, 10^{-2}, 10^{-1}\}$ and was used for $10^4$ epochs on a dataset of $10^3$ samples (RF). Initial parameters of the networks are drawn from i.i.d. gaussians: $\mathcal{N}(0, 1)$ for the teacher and $\mathcal{N}(10^{-3}, 1)$ for the student. Results are averaged over 10 different runs.

Figure 1 illustrates the behavior of the proposed algorithm (21) in a simple setting and compares it with three other methods: MMD without noise injection (blue traces), MMD with diffusion (green traces) and KSD (purple traces, [32]). Here, a student network is trained to produce the outputs of a teacher network using gradient descent. More details on the experiment are provided in Appendix G.1. As discussed in Appendix F, this setting can be seen as a *stochastic* version of the MMD flow since the kernel is estimated using random features at each iteration ((91) in Appendix G.1). Here, the MMD flow fails to converge towards the global optimum. Such behavior is consistent with the observations in [11] when the parameters are initialized from a gaussian noise with relatively high variance (which is the case here). On the other hand, adding noise to the gradient seems to lead to global convergence. Indeed, the training error decreases below $10^{-5}$ and leads to much better validation error. While adding a small diffusion term (green) help convergence, the noise-injection (red) still outperforms it. This also holds for KSD (purple) which leads to a good solution (b) although at a much higher

computational cost (a). Our noise injection method (red) is also robust to the amount of noise and achieves best performance over a wide region (c). On the other hand, MMD + diffusion (green) performs well only for much smaller values of noise that are located in a narrow region. This is expected since adding a diffusion changes the optimal solution, unlike the injection where the global optimum of the MMD remains a fixed point of the algorithm.

Another illustrative experiment on a simple flow between Gaussians is given in Appendix G.2.

## 5 Conclusion

We have introduced MMD flow, a novel flow over the space of distributions, with a practical space-time discretized implementation and a regularisation scheme to improve convergence. We provide theoretical results, highlighting intrinsic properties of the regular MMD flow, and guarantees on convergence based on recent results in optimal transport, probabilistic interpretations of PDEs, and particle algorithms. Future work will focus on a deeper understanding of regularization for MMD flow, and its application in sampling and optimization for large neural networks.

## Footnotes

[1]Also, $V_t = \nabla V + \nabla W \star \nu_t$ (see Appendix A.3) where $\star$ denotes the classical convolution.

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
