[Supplementary Material]

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

### A.1 Maximum Mean Discrepancy and Reproducing Kernel Hilbert Spaces

We recall here fundamental definitions and properties of reproducing kernel Hilbert spaces (RKHS) (see [54]) and Maximum Mean Discrepancies (MMD). Given a positive semi-definite kernel $(x, y) \mapsto k(x, y) \in \mathbb{R}$ defined for all $x, y \in \mathcal{X}$, we denote by $\mathcal{H}$ its corresponding RKHS (see [54]). The space $\mathcal{H}$ is a Hilbert space with inner product $\langle ., . \rangle_{\mathcal{H}}$ and corresponding norm $\|.\|_{\mathcal{H}}$. A key property of $\mathcal{H}$ is the reproducing property: for all $f \in \mathcal{H}, f(x) = \langle f, k(x, .) \rangle_{\mathcal{H}}$. Moreover, if $k$ is $m$-times differentiable w.r.t. each of its coordinates, then any $f \in \mathcal{H}$ is $m$-times differentiable and $\partial^{\alpha} f(x) = \langle f, \partial^{\alpha} k(x, .) \rangle_{\mathcal{H}}$ where $\alpha$ is any multi-index with $\alpha \leq m$ [56, Lemma 4.34]. When $k$ has at most quadratic growth, then for all $\mu \in \mathcal{P}_2(\mathcal{X})$, $\int k(x, x) \, \mathrm{d}\mu(x) < \infty$. In that case, for any $\mu \in \mathcal{P}_2(\mathcal{X})$, $\phi_{\mu} := \int k(., x) \, \mathrm{d}\mu(x)$ is a well defined element in $\mathcal{H}$ called the mean embedding of $\mu$. The kernel $k$ is said to be characteristic when such mean embedding is injective, that is any mean embedding is associated to a unique probability distribution. When $k$ is characteristic, it is possible to define a distance between distributions in $\mathcal{P}_2(\mathcal{X})$ called the Maximum Mean Discrepancy:

$$MMD(\mu, \nu) = \|\phi_{\mu} - \phi_{\nu}\|_{\mathcal{H}} \qquad \forall \, \mu, \nu \in \mathcal{P}_2(\mathcal{X}). \tag{22}$$

The difference between the mean embeddings of $\mu$ and $\nu$ is an element in $\mathcal{H}$ called the unnormalised witness function between $\mu$ and $\nu$: $f_{\mu,\nu} = \phi_{\nu} - \phi_{\mu}$. The MMD can also be seen as an *Integral Probability Metric*:

$$MMD(\mu, \nu) = \sup_{g \in \mathcal{B}} \int g \, \mathrm{d}\mu - \int g \, \mathrm{d}\nu \tag{23}$$

where $\mathcal{B} = \{g \in \mathcal{H} : \|g\|_{\mathcal{H}} \leq 1\}$ is the unit ball in the RKHS.

### A.2 2-Wasserstein geometry

For two given probability distributions $\nu$ and $\mu$ in $\mathcal{P}_2(\mathcal{X})$, we denote by $\Pi(\nu, \mu)$ the set of possible couplings between $\nu$ and $\mu$. In other words $\Pi(\nu, \mu)$ contains all possible distributions $\pi$ on $\mathcal{X} \times \mathcal{X}$ such that if $(X, Y) \sim \pi$ then $X \sim \nu$ and $Y \sim \mu$. The 2-Wasserstein distance on $\mathcal{P}_2(\mathcal{X})$ is defined by means of an optimal coupling between $\nu$ and $\mu$ in the following way:

$$W_2^2(\nu, \mu) := \inf_{\pi \in \Pi(\nu, \mu)} \int \|x - y\|^2 \, \mathrm{d}\pi(x, y) \qquad \forall \nu, \mu \in \mathcal{P}_2(\mathcal{X}) \tag{24}$$

It is a well established fact that such optimal coupling $\pi^*$ exists [49, 58] . Moreover, it can be used to define a path $(\rho_t)_{t \in [0,1]}$ between $\nu$ and $\mu$ in $\mathcal{P}_2(\mathcal{X})$. For a given time $t$ in $[0, 1]$ and given a sample $(x, y)$ from $\pi^*$, it is possible to construct a sample $z_t$ from $\rho_t$ by taking the convex combination of $x$ and $y$: $z_t = s_t(x, y)$ where $s_t$ is given by:

$$s_t(x, y) = (1 - t)x + ty \qquad \forall x, y \in \mathcal{X}, \ \forall t \in [0, 1]. \tag{25}$$

The function $s_t$ is well defined since $\mathcal{X}$ is a convex set. More formally, $\rho_t$ can be written as the projection or push-forward of the optimal coupling $\pi^*$ by $s_t$:

$$\rho_t = (s_t)_{\#}\pi^* \tag{26}$$

We recall that for any $T : \mathcal{X} \to \mathcal{X}$ a measurable map, and any $\rho \in \mathcal{P}(\mathcal{X})$, the push-forward measure $T_{\#}\rho$ is characterized by:

$$\int_{y \in \mathcal{X}} \phi(y) \, \mathrm{d}T_{\#}\rho(y) = \int_{x \in \mathcal{X}} \phi(T(x)) \, \mathrm{d}\rho(x) \text{ for every measurable and bounded function } \phi. \quad (27)$$

It is easy to see that (26) satisfies the following boundary conditions at $t = 0, 1$:

$$\rho_0 = \nu \qquad \rho_1 = \mu. \quad (28)$$

Paths of the form of (26) are called *displacement geodesics*. They can be seen as the shortest paths from $\nu$ to $\mu$ in terms of mass transport ([49] Theorem 5.27). It can be shown that there exists a *velocity vector field* $(t, x) \mapsto V_t(x)$ with values in $\mathbb{R}^d$ such that $\rho_t$ satisfies the continuity equation:

$$\partial_t \rho_t + div(\rho_t V_t) = 0 \qquad \forall t \in [0, 1]. \quad (29)$$

This equation expresses two facts, the first one is that $-div(\rho_t V_t)$ reflects the infinitesimal changes in $\rho_t$ as dictated by the vector field (also referred to as velocity field) $V_t$, the second one is that the total mass of $\rho_t$ does not vary in time as a consequence of the divergence theorem. Equation (29) is well defined in the distribution sense even when $\rho_t$ does not have a density. At each time $t$, $V_t$ can be interpreted as a tangent vector to the curve $(\rho_t)_{t \in [0,1]}$ so that the length $l((\rho_t)_{t \in [0,1]})$ of the curve $(\rho_t)_{t \in [0,1]}$ would be given by:

$$l((\rho_t)_{t \in [0,1]})^2 = \int_0^1 \|V_t\|_{L_2(\rho_t)}^2 \, \mathrm{d}t \quad \text{where} \quad \|V_t\|_{L_2(\rho_t)}^2 = \int \|V_t(x)\|^2 \, \mathrm{d}\rho_t(x) \quad (30)$$

This perspective allows to provide a dynamical interpretation of the $W_2$ as the length of the shortest path from $\nu$ to $\mu$ and is summarized by the celebrated Benamou-Brenier formula ([6]):

$$W_2(\nu, \mu) = \inf_{(\rho_t, V_t)_{t \in [0,1]}} l((\rho_t)_{t \in [0,1]}) \quad (31)$$

where the infimum is taken over all couples $\rho$ and $v$ satisfying (29) with boundary conditions given by (28). If $(\rho_t, V_t)_{t \in [0,1]}$ satisfies (29) and (28) and realizes the infimum in (31), it is then simply called a geodesic between $\nu$ and $\mu$; moreover it is called a constant-speed geodesic if, in addition, the norm of $V_t$ is constant for all $t \in [0, 1]$. As a consequence, (26) is a constant-speed displacement geodesic.

**Remark 1.** *Such paths should not be confused with another kind of paths called mixture geodesics. The mixture geodesic $(m_t)_{t \in [0,1]}$ from $\nu$ to $\mu$ is obtained by first choosing either $\nu$ or $\mu$ according to a Bernoulli distribution of parameter $t$ and then sampling from the chosen distribution:*

$$m_t = (1 - t)\nu + t\mu \qquad \forall t \in [0, 1]. \quad (32)$$

*Paths of the form (32) can be thought as the shortest paths between two distributions when distances on $\mathcal{P}_2(\mathcal{X})$ are measured using the MMD (see [9] Theorem 5.3). We refer to [9] for an overview of the notion of shortest paths in probability spaces and for the differences between mixture geodesics and displacement geodesics. Although, we will be interested in the MMD as a loss function, we will not consider the geodesics that are naturally associated to it and will rather consider the displacement geodesics defined in (26) for reasons that will become clear in Appendix A.4.*

### A.3 Gradient flows on the space of probability measures

Consider a real valued functional $\mathcal{F}$ defined over $\mathcal{P}_2(\mathbf{x})$. We call $\frac{\partial \mathcal{F}}{\partial \nu}$ if it exists, the unique (up to additive constants) function such that $\frac{d}{d\epsilon}\mathcal{F}(\nu + \epsilon(\nu' - \nu))|_{\epsilon=0} = \int \frac{\partial \mathcal{F}}{\partial \nu}(\nu)(\mathrm{d}\nu' - \mathrm{d}\nu)$ for any $\nu' \in \mathcal{P}_2(\mathcal{X})$. The function $\frac{\partial \mathcal{F}}{\partial \nu}$ is called the first variation of $\mathcal{F}$ evaluated at $\nu$. We consider here functionals $\mathcal{F}$ of the form:

$$\mathcal{F}(\nu) = \int U(\nu(x))\nu(x)dx + \int V(x)\nu(x)dx + \int W(x, y)\nu(x)\nu(y)dxdy \quad (33)$$

where $U$ is the internal potential, $V$ an external potential and $W$ an interaction potential. The formal gradient flow equation associated to such functional can be written (see [11], Lemma 8 to 10):

$$\frac{\partial \nu}{\partial t} = div(\nu \nabla \frac{\partial \mathcal{F}}{\partial \nu}) = div(\nu \nabla (U'(\nu) + V + W * \nu)) \quad (34)$$

where $div$ is the divergence operator and $\nabla \frac{\partial \mathcal{F}}{\partial \nu}$ is the strong subdifferential of $\mathcal{F}$ associated to the $W_2$ metric (see [1], Lemma 10.4.1). Indeed, for some generalized notion of gradient $\nabla_{W_2}$, and for sufficiently regular $\nu$ and $\mathcal{F}$, the r.h.s. of (34) can be formally written as $-\nabla_{W_2}\mathcal{F}(\nu)$. The dissipation of energy along the flow is then given by:

$$\frac{d\mathcal{F}(\nu_t)}{dt} = -D(\nu_t) \quad \text{with } D(\nu) = \int \|\nabla \frac{\partial \mathcal{F}(\nu_t(x))}{\partial \nu}\|^2 \nu_t(x)dx \tag{35}$$

Such expression can be obtained by the following formal calculations:

$$\frac{d\mathcal{F}(\nu_t)}{dt} = \int \frac{\partial \mathcal{F}(\nu_t)}{\partial \nu_t} \frac{\partial \nu_t}{\partial t} = \int \frac{\partial \mathcal{F}(\nu_t)}{\partial \nu} div(\nu_t \nabla \frac{\partial \mathcal{F}(\nu_t)}{\partial \nu}) = -\int \|\nabla \frac{\partial \mathcal{F}(\nu_t)}{\partial \nu}\|^2 d\nu_t.$$

### A.4 Displacement convexity

Just as for Euclidian spaces, an important criterion to characterize the convergence of the Wasserstein gradient flow of a functional $\mathcal{F}$ is given by displacement convexity (see [60, Definition 16.5 (1st bullet point)])):

**Definition 2.** *[Displacement convexity] We say that a functional $\nu \mapsto \mathcal{F}(\nu)$ is displacement convex if for any $\nu$ and $\nu'$ and a constant speed geodesic $(\rho_t)_{t \in [0,1]}$ between $\nu$ and $\nu'$ with velocity vector field $(V_t)_{t \in [0,1]}$ as defined by (29), the following holds:*

$$\mathcal{F}(\rho_t) \leq (1-t)\mathcal{F}(\nu_0) + t\mathcal{F}(\nu_1) \qquad \forall\, t \in [0,1]. \tag{36}$$

Definition 2 can be relaxed to a more general notion of convexity called $\Lambda$-displacement convexity (see [58, Definition 16.5 (3rd bullet point)]). We first define an admissible functional $\Lambda$:

**Definition 3.** *[Admissible $\Lambda$ functional] Consider a functional $(\rho, v) \mapsto \Lambda(\rho, v) \in \mathbb{R}$ defined for any probability distribution $\rho \in \mathcal{P}_2(\mathcal{X})$ and any square integrable vector field $v$ w.r.t $\rho$. We say that $\Lambda$ is admissible, if it satisfies:*

- *For any $\rho \in \mathcal{P}_2(\mathcal{X})$, $v \mapsto \Lambda(\rho, v)$ is a quadratic form.*
- *For any geodesic $(\rho_t)_{0 \leq t \leq 1}$ between two distributions $\nu$ and $\nu'$ with corresponding vector fields $(V_t)_{t \in [0,1]}$ it holds that $\inf_{0 \leq t \leq 1} \Lambda(\rho_t, V_t)/\|V_t\|^2_{L_2(\rho_t)} > -\infty$*

We can now define the notion of $\Lambda$-convexity:

**Definition 4.** *[$\Lambda$ convexity] We say that a functional $\nu \mapsto \mathcal{F}(\nu)$ is $\Lambda$-convex if for any $\nu, \nu' \in \mathcal{P}_2(\mathcal{X})^2$ and a constant speed geodesic $(\rho_t)_{t \in [0,1]}$ between $\nu$ and $\nu'$ with velocity vector field $(V_t)_{t \in [0,1]}$ as defined by (29), the following holds:*

$$\mathcal{F}(\rho_t) \leq (1-t)\mathcal{F}(\nu_0) + t\mathcal{F}(\nu_1) - \int_0^1 \Lambda(\rho_s, V_s)G(s,t)ds \qquad \forall\, t \in [0,1]. \tag{37}$$

*where $(\rho, v) \mapsto \Lambda(\rho, v)$ satisfies Definition 3, and $G(s,t) = s(1-t)\mathbb{I}\{s \leq t\} + t(1-s)\mathbb{I}\{s \geq t\}$. A particular case is when $\Lambda(\rho, v) = \lambda \int \|v(x)\|^2 \, d\rho(x)$ for some $\lambda \in \mathbb{R}$. In that case, (37) becomes:*

$$\mathcal{F}(\rho_t) \leq (1-t)\mathcal{F}(\nu_0) + t\mathcal{F}(\nu_1) - \frac{\lambda}{2}t(1-t)W_2^2(\nu_0, \nu_1) \qquad \forall\, t \in [0,1]. \tag{38}$$

Definition 2 is a particular case of Definition 4, where in (38) one has $\lambda = 0$.

### A.5 Comparison with the Kullback Leilber divergence flow

*Continuity equation and McKean Vlasov process.* A famous example of a free energy (33) is the Kullback-Leibler divergence, defined for $\nu, \mu \in \mathcal{P}(\mathcal{X})$ by $KL(\nu, \mu) = \int log(\frac{\nu(x)}{\mu(x)})\nu(x)dx$. Indeed, $KL(\nu, \mu) = \int U(\nu(x))dx + \int V(x)\nu(x)dx$ with $U(s) = s \log(s)$ the entropy function and $V(x) = -log(\mu(x))$. In this case, $\nabla \frac{\partial \mathcal{F}}{\partial \nu} = \nabla \log(\nu) + \nabla V = \nabla \log(\frac{\nu}{\mu})$ and equation (34) leads to the classical Fokker-Planck equation

$$\frac{\partial \nu}{\partial t} = div(\nu \nabla V) + \Delta \nu, \tag{39}$$

where $\Delta$ is the Laplacian operator. It is well-known (see for instance [29]) that the distribution of the Langevin diffusion in (40) satisfies (39),

$$dX_t = -\nabla \log \mu(X_t) dt + \sqrt{2} dB_t. \tag{40}$$

Here, $(B_t)_{t \geq 0}$ is a $d$-dimensional Brownian motion. While the entropy term in the $KL$ functional prevents the particles from "crashing" onto the mode of $\mu$, this role could be played by the interaction energy $W$ defined in (4) for the MMD. Indeed, consider for instance the gaussian kernel $k(x, x') = e^{-\|x-x'\|^2}$. It is convex thus attractive at long distances ($\|x - x'\| > 1$) but repulsive at small distances so repulsive.

*Convergence to a global minimum.* The solution to the Fokker-Planck equation describing the gradient flow of the $KL$ can be shown to converge towards $\mu$ under mild assumptions. This follows from the displacement convexity of the $KL$ along the Wasserstein geodesics. Unfortunately the MMD is not displacement convex in general, as shown in Section 3.1 or Appendix D.2. This makes the task of proving the convergence of the gradient flow of the MMD to the global optimum $\mu$ much harder.

*Sampling algorithms derived from gradient flows.* Two settings are usually encountered in the sampling literature: *density-based*, i.e. the target $\mu$ is known up to a constant, or *sample-based*, i.e. only a set of samples $X \sim \mu$ is accessible. The Unadjusted Langevin Algorithm (ULA), which involves a time-discretized version of the Langevin diffusion falls into the first category since it requires the knowledge of $\nabla \log \mu$. In a sample-based setting, it may be difficult to adapt the ULA algorithm, since this would require to estimate $\nabla \log(\mu)$ based on a set of samples of $\mu$, before plugging this estimate in the update of the algorithm. This problem, sometimes referred to as *score estimation* in the literature, has been the subject of a lot of work but remains hard especially in high dimensions (see [57],[35],[51]). In contrast, the discretized flow (in time and space) of the MMD presented in Section 4.2 is naturally adapted to the sample-based setting.

# B    Main assumptions

We state here all the assumptions on the kernel $k$ used to prove all the results:

(A) $k$ is continuously differentiable on $\mathcal{X}$ with $L$-Lipschitz gradient: $\|\nabla k(x, x') - \nabla k(y, y')\| \leq L(\|x - y\| + \|x' - y'\|)$ for all $x, x', y, y' \in \mathcal{X}$.

(B) $k$ is twice differentiable on $\mathcal{X}$.

(C) $\|Dk(x, y)\| \leq \lambda$ for all $x, y \in \mathcal{X}$, where $Dk(x, y)$ is an $\mathbb{R}^{d^2} \times \mathbb{R}^{d^2}$ matrix with entries given by $\partial_{x_i} \partial_{x_j} \partial_{x'_i} \partial_{x'_j} k(x, y)$.

(D) $\sum_{i=1}^{d} \|\partial_i k(x, .) - \partial_i k(y, .)\|_{\mathcal{H}}^2 \leq \lambda^2 \|x - y\|^2$ for all $x, y \in \mathcal{X}$.

# C    Construction of the gradient flow of the MMD

## C.1    Continuous time flow

Existence and uniqueness of a solution to (5) and (6) is guaranteed under Lipschitz regularity of $\nabla k$.

*Proof of Proposition 1.* [Existence and uniqueness] Under Assumption (A), the map $(x, \nu) \mapsto \nabla f_{\mu,\nu}(x) = \int \nabla k(x, .) d\nu - \int \nabla k(x, .) d\mu$ is Lipschitz continuous on $\mathcal{X} \times \mathcal{P}_2(\mathcal{X})$ (endowed with the product of the canonical metric on $\mathcal{X}$ and $W_2$ on $\mathcal{P}_2(\mathcal{X})$), see Proposition 21. Hence, we benefit from standard existence and uniqueness results of McKean-Vlasov processes (see [30]). Then, it is straightforward to verify that the distribution of (6) is solution of (5) by ItÃŽ's formula (see [28]). The uniqueness of the gradient flow, given a starting distribution $\nu_0$, results from the $\lambda$-convexity of $\mathcal{F}$ (for $\lambda = 3L$) which is given by Lemma 14, and [1, Theorem 11.1.4]. The existence derive from the fact that the sub-differential of $\mathcal{F}$ is single-valued, as stated by (2), and that any $\nu_0$ in $\mathcal{P}_2(\mathcal{X})$ is in the domain of $\mathcal{F}$. One can then apply [1, Theorem 11.1.6 and Corollary 11.1.8]. $\qquad \square$

*Proof of Proposition 2.* [Decay of the MMD] Recalling the discussion in Appendix A.3, the time derivative of $\mathcal{F}(\nu_t)$ along the flow is formally given by (35). But we know from (2) that the strong

differential $\nabla \frac{\delta \mathcal{F}(\nu)}{\delta \nu}$ is given by $\nabla f_{\mu,\nu}$. Therefore, one formally obtains the desired expression by exchanging the order of derivation and integration, performing an integration by parts and using the continuity equation (see (35)). We refer to [41] for similar calculations. One can also obtain directly the same result using the energy identity in [1, Theorem 11.3.2] which holds for $\lambda$-displacement convex functionals. The result applies here since, by Lemma 14, we know that $\mathcal{F}$ is $\lambda$-displacement convex with $\lambda = 3L$. $\qquad\square$

## C.2 Time-discretized flow

We prove that (8) approximates (5). To make the dependence on the step-size $\gamma$ explicit, we will write: $\nu_{n+1}^\gamma = (I - \gamma \nabla f_{\mu,\nu_n^\gamma})_\# \nu_n^\gamma$ (so $\nu_n^\gamma = \nu_n$ for any $n \geq 0$). We start by introducing an auxiliary sequence $\bar{\nu}_n^\gamma$ built by iteratively applying $\nabla f_{\mu,\nu_{\gamma n}}$ where $\nu_{\gamma n}$ is the solution of (5) at time $t = \gamma n$:

$$\bar{\nu}_{n+1}^\gamma = (I - \gamma \nabla f_{\mu,\nu_{\gamma n}})_\# \bar{\nu}_n^\gamma \tag{41}$$

with $\bar{\nu}_0 = \nu_0$. Note that the latter sequence involves the continuous-time process $\nu_t$ of (5) with $t = \gamma n$. Using $\nu_n^\gamma$, we also consider the interpolation path $\rho_t^\gamma = (I - (t - n\gamma)\nabla f_{\mu,\nu_n^\gamma})_\# \nu_n^\gamma$ for all $t \in [n\gamma, (n+1)\gamma)$ and $n \in \mathbb{N}$, which is the same as in Proposition 3.

*Proof of Proposition 3.* Let $\pi$ be an optimal coupling between $\nu_n^\gamma$ and $\nu_{\gamma n}$, and $(x, y)$ a sample from $\pi$. For $t \in [n\gamma, (n+1)\gamma)$ we write $y_t = y_{n\gamma} - \int_{n\gamma}^t \nabla f_{\mu,\nu_s}(y_u)\, \mathrm{d}u$ and $x_t = x - (t - n\gamma)\nabla f_{\mu,\nu_n^\gamma}(x)$ where $y_{n\gamma} = y$. We also introduce the approximation error $E(t, n\gamma) := y_t - y + (t - n\gamma)\nabla f_{\mu,\nu_{\gamma n}}(y)$ for which we know by Lemma 12 that $\mathcal{E}(t, n\gamma) := \mathbb{E}[E(t, n\gamma)^2]^{\frac{1}{2}}$ is upper-bounded by $(t - n\gamma)^2 C$ for some positive constant $C$ that depends only on $T$ and the Lipschitz constant $L$. This allows to write:

$$
\begin{aligned}
W_2(\rho_t^\gamma, \nu_t) &\leq \mathbb{E}\left[\left\| y - x + (t - n\gamma)(\nabla f_{\mu,\nu_n^\gamma}(x) - \nabla f_{\mu,\nu_{\gamma n}}(y)) + E(t, n\gamma) \right\|^2\right]^{\frac{1}{2}} \\
&\leq W_2(\nu_n^\gamma, \nu_{\gamma n}) + 4L(t - n\gamma)W_2(\nu_n^\gamma, \nu_{\gamma n}) + \mathcal{E}(t, n\gamma) \\
&\leq (1 + 4\gamma L)W_2(\nu_n^\gamma, \nu_{\gamma n}) + (t - \gamma n)^2 C \\
&\leq (1 + 4\gamma L)\left(W_2(\nu_n^\gamma, \bar{\nu}_n^\gamma) + W_2(\nu_{\gamma n}, \bar{\nu}_n^\gamma)\right) + \gamma^2 C \\
&\leq \gamma\left[(1 + 4\gamma L)M(T) + \gamma C\right]
\end{aligned}
$$

The second line is obtained using that $\nabla f_{\mu,\nu_{\gamma n}}(x)$ is jointly $2L$-Lipschitz in $x$ and $\nu$ (see Proposition 21) and by the fact that $W_2(\nu_n^\gamma, \nu_{\gamma n}) = \mathbb{E}_\pi[\|y - x\|^2]^{\frac{1}{2}}$. The third one is obtained using $t - n\gamma \leq \gamma$. For the last inequality, we used Lemmas 10 and 11 where $M(T)$ is a constant that depends only on $T$. Hence for $\gamma \leq \frac{1}{4L}$ we get $W_2(\rho_t^\gamma, \nu_t) \leq \gamma(\frac{C}{4L} + 2M(T))$. $\qquad\square$

**Lemma 10.** *For any $n \geq 0$:*

$$W_2(\nu_{\gamma n}, \bar{\nu}_n^\gamma) \leq \gamma \frac{C}{2L}(e^{n\gamma 2L} - 1)$$

*Proof.* Let $\pi$ be an optimal coupling between $\bar{\nu}_n^\gamma$ and $\nu_{\gamma n}$ and $(\bar{x}, x)$ a joint sample from $\pi$. Consider also the joint sample $(\bar{y}, y)$ obtained from $(\bar{x}, x)$ by applying the gradient flow of $\mathcal{F}$ in continuous time to get $y := x_{(n+1)\gamma} = x_{n\gamma} - \int_{n\gamma}^{(n+1)\gamma} \nabla f_{\mu,\nu_s}(x_u)\, \mathrm{d}u$ with $x_{n\gamma} = x$ and by taking a discrete step from $\bar{x}$ to write $\bar{y} = \bar{x} - \gamma \nabla f_{\mu,\nu_{\gamma n}}(\bar{x})$. It is easy to see that $y \sim \nu_{\gamma(n+1)}$ (i.e. a sample from the continous process (5) at time $t = (n+1)\gamma$) and $\bar{y} \sim \bar{\nu}_{n+1}^\gamma$ (i.e. a sample from (41)). Moreover, we introduce the approximation error $E((n+1)\gamma, n\gamma) := y - x + \gamma \nabla f_{\mu,\nu_{\gamma n}}(x)$ for which we know by Lemma 12 that $\mathcal{E}((n+1)\gamma, n\gamma) := \mathbb{E}[E((n+1)\gamma, n\gamma)^2]^{\frac{1}{2}}$ is upper-bounded by $\gamma^2 C$ for some positive constant $C$ that depends only on $T$ and the Lipschitz constant $L$. Denoting by $a_n = W_2(\nu_{\gamma n}, \bar{\nu}_n^\gamma)$, one can therefore write:

$$
\begin{aligned}
a_{n+1} &\leq \mathbb{E}_\pi\left[\left\| x - \gamma \nabla f_{\mu,\nu_{\gamma n}}(x) - \bar{x} + \gamma \nabla f_{\mu,\nu_{\gamma n}}(\bar{x}) + E((n+1)\gamma, n\gamma) \right\|^2\right]^{\frac{1}{2}} \\
&\leq \mathbb{E}_\pi\left[\|x - \bar{x}\|^2\right]^{\frac{1}{2}} + \gamma \mathbb{E}_\pi\left[\left\| \nabla f_{\mu,\nu_{\gamma n}}(x) - \nabla f_{\mu,\nu_{\gamma n}}(\bar{x})) \right\|^2\right]^{\frac{1}{2}} + \gamma^2 C
\end{aligned}
$$

Using that $\nabla f_{\mu,\nu_{\gamma n}}$ is $2L$-Lipschitz by Proposition 21 and recalling that $\mathbb{E}_{\pi}\left[\|x - \bar{x}\|^2\right]^{\frac{1}{2}} = W_2(\nu_{\gamma n}, \bar{\nu}_n^{\gamma})$, we get the recursive inequality $a_{n+1} \leq (1 + 2\gamma L)a_n + \gamma^2 C$. Finally, using Lemma 26 and recalling that $a_0 = 0$, since by definition $\bar{\nu}_0^{\gamma} = \nu_0^{\gamma}$, we conclude that $a_n \leq \gamma\frac{C}{2L}(e^{n\gamma 2L} - 1)$. $\qquad\square$

**Lemma 11.** *For any $T > 0$ and $n$ such that $n\gamma \leq T$*

$$W_2(\nu_n^{\gamma}, \bar{\nu}_n^{\gamma}) \leq \gamma\frac{C}{8L^2}(e^{4TL} - 1)^2 \tag{42}$$

*Proof.* Consider now an optimal coupling $\pi$ between $\bar{\nu}_n^{\gamma}$ and $\nu_n^{\gamma}$. Similarly to Lemma 10, we denote by $(\bar{x}, x)$ a joint sample from $\pi$ and $(\bar{y}, y)$ is obtained from $(\bar{x}, x)$ by applying the discrete updates : $\bar{y} = \bar{x} - \gamma\nabla f_{\mu,\nu_{\gamma n}}(\bar{x})$ and $y = x - \gamma\nabla f_{\mu,\nu_n^{\gamma}}(x)$. We again have that $y \sim \nu_{n+1}^{\gamma}$ (i.e. a sample from the time discretized process (8)) and $\bar{y} \sim \bar{\nu}_{n+1}^{\gamma}$ (i.e. a sample from (41)). Now, denoting by $b_n = W_2(\nu_n^{\gamma}, \bar{\nu}_n^{\gamma})$, it is easy to see from the definition of $\bar{y}$ and $y$ that we have:

$$b_{n+1} \leq \mathbb{E}_{\pi}\left[\left\|x - \gamma\nabla f_{\mu,\nu_n^{\gamma}}(x) - \bar{x} + \gamma\nabla f_{\mu,\nu_{\gamma n}}(\bar{x})\right\|^2\right]^{\frac{1}{2}}$$

$$\leq (1 + 2\gamma L)\mathbb{E}_{\pi}\left[\|x - \bar{x}\|^2\right]^{\frac{1}{2}} + 2\gamma L W_2(\nu_n^{\gamma}, \nu_{\gamma n}))$$

$$\leq (1 + 4\gamma L)b_n + \gamma L W_2(\bar{\nu}_n^{\gamma}, \nu_{\gamma n})$$

The second line is obtained recalling that $\nabla f_{\mu,\nu}(x)$ is $2L$-Lipschitz in both $x$ and $\nu$ by Proposition 21. The third line follows by triangular inequality and using $\mathbb{E}_{\pi}\left[\|x - \bar{x}\|^2\right]^{\frac{1}{2}} = W_2(\nu_n^{\gamma}, \bar{\nu}_n^{\gamma}) = b_n$, since $\pi$ is an optimal coupling between $\bar{\nu}_n^{\gamma}$ and $\nu_n^{\gamma}$. By Lemma 10, we have $W_2(\bar{\nu}_n^{\gamma}, \nu_{\gamma n}) \leq \gamma\frac{C}{2L}(e^{2n\gamma L} - 1)$, hence, for any $n$ such that $n\gamma \leq T$ we get the recursive inequality

$$b_{n+1} \leq (1 + 4\gamma L)b_n + (C/2L)\gamma^2(e^{2TL} - 1).$$

Finally, using again Lemma 26, it follows that $b_n \leq \gamma\frac{C}{8L^2}(e^{4TL} - 1)^2$. $\qquad\square$

**Lemma 12.** *[Taylor expansion] Consider the process $\dot{x}_t = -\nabla f_{\mu,\nu_t}(x_t)$, and denote by $\mathcal{E}(t, s) = \mathbb{E}[\|x_t - x_s + (t - s)\nabla f_{\mu,\nu_s}(x_s)\|^2]^{\frac{1}{2}}$ for $0 \leq s \leq t \leq T$. Then one has:*

$$\mathcal{E}(t, s) \leq 2L^2 r_0 e^{LT}(t - s)^2 \tag{43}$$

*with $r_0 = \mathbb{E}_{(x,z)\sim\nu_0\otimes\mu}[\|x - z\|]$*

*Proof.* By definition of $x_t$ and $\mathcal{E}(t, s)$ one can write:

$$\mathcal{E}(t, s) = \mathbb{E}\left[\left\|\int_s^t (\nabla f_{\mu,\nu_s}(x_s) - \nabla f_{\mu,\nu_u}(x_u))\, du\right\|^2\right]^{\frac{1}{2}}$$

$$\leq \int_s^t \mathbb{E}\left[\|(\nabla f_{\mu,\nu_s}(x_s) - \nabla f_{\mu,\nu_u}(x_u))\|^2\right]^{\frac{1}{2}} du$$

$$\leq 2L \int_s^t \mathbb{E}\left[(\|x_s - x_u\| + W_2(\nu_s, \nu_u))^2\right]^{\frac{1}{2}} du \leq 4L \int_s^t \mathbb{E}\left[\|x_s - x_u\|^2\right]^{\frac{1}{2}} du$$

Where we used an integral expression for $x_t$ in the first line then applied a triangular inequality for the second line. The last line is obtained recalling that $\nabla f_{\mu,\nu}(x)$ is jointly $2L$-Lipschitz in $x$ and $\nu$ by Proposition 21 and that $W_2(\nu_s, \nu_u) \leq \mathbb{E}\left[\|x_s - x_u\|^2\right]^{\frac{1}{2}}$. Now we use again an integral expression for $x_u$ which further gives:

$$\mathcal{E}(t, s) \leq 4L \int_s^t \mathbb{E}\left[\left\|\int_s^u \nabla f_{\mu,\nu_l}(x_l)\, dl\right\|^2\right]^{\frac{1}{2}} du$$

$$\leq 4L \int_s^t \int_s^u \mathbb{E}\left[\|\mathbb{E}\left[\nabla_1 k(x_l, x_l') - \nabla_1 k(x_l, z)\right]\|^2\right]^{\frac{1}{2}} dl\, du$$

$$\leq 4L^2 \int_s^t \int_s^u \mathbb{E}\left[\|x_l' - z\|\right] dl\, du$$

Again, the second line is obtained using a triangular inequality and recalling the expression of $\nabla f_{\mu,\nu}(x)$ from Proposition 21. The last line uses that $\nabla k$ is $L$-Lipschitz by Assumption (A). Now we need to make sure that $\|x'_l - z\|$ remains bounded at finite times. For this we will first show that $r_t = \mathbb{E}[\|x_t - z\|]$ satisfies an integro-differential inequality:

$$r_t \leq \mathbb{E}\left[\left\|x_0 - z - \int_0^t \nabla f_{\mu,\nu_s}(x_s)\,\mathrm{d}s\right\|\right]$$

$$\leq r_0 + \int_0^t \mathbb{E}\left[\|\nabla_1 k(x_s, x'_s) - \nabla_1 k(x_s, z)\|\right]\mathrm{d}s \leq r_0 + L \int_0^t r_s\,\mathrm{d}s$$

Again, we used an integral expression for $x_t$ in the first line, then a triangular inequality recalling the expression of $\nabla f_{\mu,\nu_s}$. The last line uses again that $\nabla k$ is $L$-Lipschitz. By Gronwall's lemma it is easy to see that $r_t \leq r_0 e^{Lt}$ at all times. Moreover, for all $t \leq T$ we have a fortiori that $r_t \leq r_0 e^{LT}$. Recalling back the upper-bound on $\mathcal{E}(t,s)$ we have finally:

$$\mathcal{E}(t,s) \leq 4L^2 r_0 e^{LT} \int_s^t \int_s^u \mathrm{d}l\,\mathrm{d}u = 2L^2 r_0 e^{LT}(t-s)^2$$

$\square$

We show now that (8) decreases the functional $\mathcal{F}$. In all the proofs, the step-size $\gamma$ is fixed.

*Proof of Proposition 4.* Consider a path between $\nu_n$ and $\nu_{n+1}$ of the form $\rho_t = (I - \gamma t \nabla f_{\mu,\nu_n})_{\#}\nu_n$. We know by Proposition 21 that $\nabla f_{\mu,\nu_n}$ is $2L$ Lipschitz, thus by Lemma 22 and using $\phi(x) = -\gamma \nabla f_{\mu,\nu_n}(x)$, $\psi(x) = x$ and $q = \nu_n$ it follows that $\mathcal{F}(\rho_t)$ is differentiable and hence absolutely continuous. Therefore one can write:

$$\mathcal{F}(\rho_1) - \mathcal{F}(\rho_0) = \dot{\mathcal{F}}(\rho_0) + \int_0^1 \dot{\mathcal{F}}(\rho_t) - \dot{\mathcal{F}}(\rho_0)dt. \tag{44}$$

Moreover, Lemma 22 also allows to write:

$$\dot{\mathcal{F}}(\rho_0) = -\gamma \int \|\nabla f_{\mu,\nu_n}(x)\|^2 d\nu_n(x); \qquad |\dot{\mathcal{F}}(\rho_t) - \dot{\mathcal{F}}(\rho_0)| \leq 3Lt\gamma^2 \int \|\nabla f_{\mu,\nu_n}(X)\|^2 d\nu_n(X).$$

where $t \leq 1$. Hence, the result follows directly by applying the above expression to (44). $\square$

# D  Convergence of the gradient flow of the MMD

## D.1  Equilibrium condition

We discuss here the equilibrium condition (11) and relate it to [41, Assumption A]. Recall that (11) is given by: $\int \|\nabla f_{\mu,\nu^*}(x)\|^2\,\mathrm{d}\nu^*(x) = 0$. Under some mild assumptions on the kernel which are states in [41, Appendix C.1] it is possible to write (11) as:

$$\int \|\nabla f_{\mu,\nu^*}(x)\|^2\,\mathrm{d}\nu^*(x) = \langle f_{\mu,\nu^*}, D_{\nu^*} f_{\mu,\nu^*}\rangle_{\mathcal{H}} = 0$$

where $D_{\nu^*}$ is a Hilbert-Schmidt operator given by:

$$D_{\nu^*} = \int \sum_{i=1}^d \partial_i k(x,.) \otimes \partial_i k(x,.)\,\mathrm{d}\nu^*(x)$$

Hence (11) is equivalent to say that $f_{\mu,\nu^*}$ belongs to the null space of $D_{\nu^*}$. In [41, Theorem 2], a similar equilibrium condition is derived by considering the time derivative of the MMD along the KSD gradient flow:

$$\frac{1}{2}\frac{d}{dt}MMD^2(\mu,\nu_t) = -\lambda\langle f_{\mu,\nu_t}, (\frac{1}{\lambda}I - (D_{\nu_t} + \lambda I)^{-1})f_{\mu,\nu_t}\rangle_{\mathcal{H}}$$

The r.h.s is shown to be always negative and thus the MMD decreases in time. Hence, as $t$ approaches $\infty$, the r.h.s tends to 0 since the MMD converges to some limit value $l$. This provides the equilibrium condition:

$$\lambda\langle f_{\mu,\nu^*}, (\frac{1}{\lambda}I - (D_{\nu^*} + \lambda I)^{-1})f_{\mu,\nu^*}\rangle_{\mathcal{H}} = 0$$

It is further shown in [41, Lemma 2] that the above equation is also equivalent to having $f_{\mu,\nu^*}$ in the null space of $D_{\nu^*}$ in the case when $D_{\nu^*}$ has finite dimensions. We generalize this statement to infinite dimension in Proposition 13. In [41, Assumption A], it is simply assumed that if $f_{\mu,\nu^*} \neq 0$ then $D_{\nu^*}f_{\mu,\nu^*} \neq 0$ which exactly amounts to assuming that local optima which are not global don't exist.

**Proposition 13.**

$$\langle f_{\mu,\nu^*}, (\frac{1}{\lambda}I - (D_{\nu^*} + \lambda I)^{-1})f_{\mu,\nu*}\rangle_{\mathcal{H}} = 0 \iff f_{\mu,\nu^*} \in null(D_{\nu^*})$$

*Proof.* This follows simply by recalling $D_{\nu^*}$ is a symmetric non-negative Hilbert-Schmidt operator it has therefore an eigen-decomposition of the form:

$$D_{\nu^*} = \sum_{i=1}^{\infty} \lambda_i e_i \otimes e_i$$

where $e_i$ is an ortho-norrmal basis of $\mathcal{H}$ and $\lambda_i$ are non-negative. Moreover, $f_{\mu,\nu^*}$ can be decomposed in $(e_i)_{1 \leq i}$ in the form:

$$f_{\mu,\nu^*} = \sum_{i=0}^{\infty} \alpha_i e_i$$

where $\alpha_i$ is a squared integrable sequence. It follows that $\langle f_{\mu,\nu^*}, (\frac{1}{\lambda}I - (D_{\nu^*} + \lambda I)^{-1})f_{\mu,\nu*}\rangle_{\mathcal{H}}$ can be written as:

$$\langle f_{\mu,\nu^*}, (\frac{1}{\lambda}I - (D_{\nu^*} + \lambda I)^{-1})f_{\mu,\nu*}\rangle_{\mathcal{H}} = \sum_{i=1}^{\infty} \frac{\lambda_i}{\lambda_i + \lambda}\alpha_i^2$$

Hence, if $f_{\mu,\nu^*} \in null(D_{\nu^*})$ then $\langle f_{\mu,\nu^*}, D_{\nu^*}f_{\mu,\nu^*}\rangle_{\mathcal{H}} = 0$, so that $\sum_{i=1}^{\infty} \lambda_i \alpha_i^2 = 0$. Since $\lambda_i$ are non-negative, this implies that $\lambda_i \alpha_i^2 = 0$ for all $i$. Therefore, it must be that $\langle f_{\mu,\nu^*}, (\frac{1}{\lambda}I - (D_{\nu^*} + \lambda I)^{-1})f_{\mu,\nu*}\rangle_{\mathcal{H}} = 0$. Similarly, if $\langle f_{\mu,\nu^*}, (\frac{1}{\lambda}I - (D_{\nu^*} + \lambda I)^{-1})f_{\mu,\nu*}\rangle_{\mathcal{H}} = 0$ then $\frac{\lambda_i \alpha_i^2}{\lambda_i + \lambda} = 0$ hence $\langle f_{\mu,\nu^*}, D_{\nu^*}f_{\mu,\nu*}\rangle_{\mathcal{H}} = 0$. This means that $f_{\mu,\nu^*}$ belongs to $null(D_{\nu^*})$. $\qquad\square$

### D.2 $\Lambda$-displacement convexity of the MMD

We provide now a proof of Proposition 5:

*Proof of Proposition 5.* [$\Lambda$- displacement convexity of the MMD] To prove that $\nu \mapsto \mathcal{F}(\nu)$ is $\Lambda$-convex we need to compute the second time derivative $\ddot{\mathcal{F}}(\rho_t)$ where $(\rho_t)_{t \in [0,1]}$ is a displacement geodesic between two probability distributions $\nu_0$ and $\nu_1$ as defined in (26). Such geodesic always exists and can be written as $\rho_t = (s_t)_{\#}\pi$ with $s_t = x + t(y - x)$ for all $t \in [0, 1]$ and $\pi$ is an optimal coupling between $\nu_0$ and $\nu_1$ ([49], Theorem 5.27). We denote by $V_t$ the corresponding velocity vector as defined in (29). Recall that $\mathcal{F}(\rho_t) = \frac{1}{2}\|f_{\mu,\rho_t}\|_{\mathcal{H}}^2$, with $f_{\mu,\rho_t}$ defined in (1). We start by computing the first derivative of $t \mapsto \mathcal{F}(\rho_t)$. Since Assumptions (A) and (B) hold, Lemma 23 applies for $\phi(x,y) = y - x$, $\psi(x,y) = x$ and $q = \pi$, thus we know that $\ddot{\mathcal{F}}(\rho_t)$ is well defined and given by:

$$\begin{aligned}
\ddot{\mathcal{F}}(\rho_t) =& \mathbb{E}\left[(y-x)^T \nabla_1 \nabla_2 k(s_t(x,y), s_t(x',y'))(y'-x')\right] \\
&+ \mathbb{E}\left[(y-x)^T (H_1 k(s_t(x,y), s_t(x',y')) - H_1 k(s_t(x,y), z))(y-x)\right]
\end{aligned} \tag{45}$$

Moreover, Assumption (C) also holds which means by Lemma 23 that the second term in (45) can be lower-bounded by $-\sqrt{2}\lambda d\mathcal{F}(\rho_t)\mathbb{E}[\|y-x\|^2]$ so that:

$$\ddot{\mathcal{F}}(\rho_t) = \mathbb{E}\left[(y-x)^T \nabla_1 \nabla_2 k(s_t(x,y), s_t(x',y'))(y'-x')\right] - \sqrt{2}\lambda d\mathcal{F}(\rho_t)\mathbb{E}[\|y-x\|^2]$$

Recall now that $(\rho_t)_{t \in [0,1]}$ is a constant speed geodesic with velocity vector $(V_t)_{t \in [0,1]}$ thus by a change of variable, one further has:

$$\ddot{\mathcal{F}}(\rho_t) \geq \int \left[V_t^T(x)\nabla_1\nabla_2 k(x,x')V_t(x')\right] \mathrm{d}\rho_t(x) - \sqrt{2}\lambda d\mathcal{F}(\rho_t) \int \|V_t(x)\|^2 \, \mathrm{d}\rho_t(x).$$

Now we can introduce the function $\Lambda(\rho, v) = \langle v, (C_\rho - \sqrt{2}\lambda d\mathcal{F}(\rho)^{\frac{1}{2}}I)v\rangle_{L_2(\rho)}$ which is defined for any pair $(\rho, v)$ with $\rho \in \mathcal{P}_2(\mathcal{X})$ and $v$ a square integrable vector field in $L_2(\rho)$ and where $C_\rho$ is

a non-negative operator given by $(C_\rho v)(x) = \int \nabla_x \nabla_{x'} k(x,x') v(x') d\rho(x')$ for any $x \in \mathcal{X}$. This allows to write $\ddot{\mathcal{F}}(\rho_t) \geq \Lambda(\rho_t, V_t)$. It is clear that $\Lambda(\rho,.)$ is a quadratic form on $L_2(\rho)$ and satisfies the requirement in Definition 3. Finally, using Lemma 24 and Definition 4 we conclude that $\mathcal{F}$ is $\Lambda$-convex. Moreover, by the reproducing property we also know that for all $\rho \in \mathcal{P}_2(\mathcal{X})$:

$$\mathbb{E}_\rho \left[ v(x)^T \nabla_1 \nabla_2 k(x,x') v(x') \right] = \mathbb{E}_\rho \left[ \langle v(x)^T \nabla_1 k(x,.), v(x')^T \nabla_1 k(x',.) \rangle_{\mathcal{H}} \right].$$

By Bochner integrability of $v(x)^T \nabla_1 k(x,.)$ it is possible to exchange the order of the integral and the inner-product [46, Theorem 6]. This leads to the expression $\|\mathbb{E}[v(x)^T \nabla_1 k(x,.)]\|^2_{\mathcal{H}}$. Hence $\Lambda(\rho,v)$ has a second expression of the form:

$$\Lambda(\rho,v) = \left\| \mathbb{E}_\rho \left[ v(x)^T \nabla_1 k(x,.) \right] \right\|^2_{\mathcal{H}} - \sqrt{2}\lambda d\mathcal{F}(\rho)^{\frac{1}{2}} \mathbb{E}_\rho \left[ \|v(x)\|^2 \right].$$

$\square$

We also provide a result showing $\Lambda$ convexity for $\mathcal{F}$ only under Assumption (A):

**Lemma 14** ($\Lambda$-displacement convexity). *Under Assumption (A), for any $\nu, \nu' \in \mathcal{P}_2(\mathcal{X})$ and any constant speed geodesic $\rho_t$ from $\nu$ to $\nu'$, $\mathcal{F}$ satisfies for all $0 \leq t \leq 1$:*

$$\mathcal{F}(\rho_t) \leq (1-t)\mathcal{F}(\nu) + t\mathcal{F}(\nu') + 3LW_2^2(\nu, \nu')$$

*Proof.* Let $\rho_t$ be a constant speed geodesic of the form $\rho_t = s_t \# \pi$ where $\pi$ is an optimal coupling between $\nu$ and $\nu'$ and $s_t(x,y) = x + t(y-x)$. Since Assumption (A) holds, one can apply Lemma 22 with $\psi(x,y) = x$, $\phi(x,y) = y-x$ and $q = \pi$. Hence, one has that $\mathcal{F}(\rho_t)$ is differentiable and its differential satisfies:

$$|\dot{\mathcal{F}}(\rho_t) - \dot{\mathcal{F}}(\rho_s)| \leq 3L|t-s| \int \|y-x\|^2 \, \mathrm{d}\pi(x,y)$$

This implies that $\dot{\mathcal{F}}(\rho_t)$ is Lipschitz continuous and therefore is differentiable for almost all $t \in [0,1]$ by Rademacher's theorem. Hence, $\ddot{\mathcal{F}}(\rho_t)$ is well defined for almost all $t \in [0,1]$. Moreover, from the above inequality it follows that $\ddot{\mathcal{F}}(\rho_t) \geq -3L \int \|y-x\|^2 \, \mathrm{d}\pi(x,y) = -3LW_2^2(\nu, \nu')$ for almost all $t \in [0,1]$. Using Lemma 24 it follows directly that $\mathcal{F}$ satisfies the desired inequality. $\square$

### D.3 Descent up to a barrier

To provide a proof of Theorem 6, we need the following preliminary results. Firstly, an upper-bound on a scalar product involving $\nabla f_{\mu,\nu}$ for any $\mu, \nu \in \mathcal{P}_2(\mathcal{X})$ in terms of the loss functional $\mathcal{F}$, is obtained using the $\Lambda$-displacement convexity of $\mathcal{F}$ in Lemma 15. Then, an EVI (Evolution Variational Inequality) is obtained in Proposition 16 on the gradient flow of $\mathcal{F}$ in $W_2$. The proof of the theorem is given afterwards.

**Lemma 15.** *Let $\nu$ be a distribution in $\mathcal{P}_2(\mathcal{X})$ and $\mu$ the target distribution such that $\mathcal{F}(\mu) = 0$. Let $\pi$ be an optimal coupling between $\nu$ and $\mu$, and $(\rho_t)_{t \in [0,1]}$ the displacement geodesic defined by (26) with its corresponding velocity vector $(V_t)_{t \in [0,1]}$ as defined in (29). Finally let $\nabla f_{\nu,\mu}(X)$ be the gradient of the unnormalised witness function between $\mu$ and $\nu$. The following inequality holds:*

$$\int \nabla f_{\mu,\nu}(x).(y-x)d\pi(x,y) \leq \mathcal{F}(\mu) - \mathcal{F}(\nu) - \int_0^1 \Lambda(\rho_s, V_s)(1-s)ds$$

*where $\Lambda$ is defined Proposition 5.*

*Proof.* Recall that for all $t \in [0,1]$, $\rho_t$ is given by $\rho_t = (s_t)_{\#}\pi$ with $s_t = x + t(y-x)$. By $\Lambda$-convexity of $\mathcal{F}$ the following inequality holds:

$$\mathcal{F}(\rho_t) \leq (1-t)\mathcal{F}(\nu) + t\mathcal{F}(\mu) - \int_0^1 \Lambda(\rho_s, V_s)G(s,t)ds$$

Hence by bringing $\mathcal{F}(\nu)$ to the l.h.s and dividing by $t$ and then taking its limit at 0 it follows that:

$$\dot{\mathcal{F}}(\rho_t)|_{t=0} \leq \mathcal{F}(\mu) - \mathcal{F}(\nu) - \int_0^1 \Lambda(\rho_s, V_s)(1-s)ds. \tag{46}$$

where $\dot{\mathcal{F}}(\rho_t) = d\mathcal{F}(\rho_t)/dt$ and since $\lim_{t \to 0} G(s,t) = (1-s)$. Moreover, under Assumption (A), Lemma 22 applies for $\phi(x,y) = y - x$, $\psi(x,y) = x$ and $q = \pi$. It follows therefore that $\dot{\mathcal{F}}(\rho_t)$ is differentiable with time derivative given by: $\dot{\mathcal{F}}(\rho_t) = \int \nabla f_{\mu,\rho_t}(s_t(x,y)).(y-x)\,d\pi(x,y)$. Hence at $t = 0$ we get: $\dot{\mathcal{F}}(\rho_t)|_{t=0} = \int \nabla f_{\mu,\nu}(x).(y-x)\,d\pi(x,y)$ which shows the desired result when used in (46). $\qquad\square$

**Proposition 16.** *Consider the sequence of distributions $\nu_n$ obtained from (8). For $n \geq 0$, consider the scalar $K(\rho^n) := \int_0^1 \Lambda(\rho_s^n, V_s^n)(1-s)\,ds$ where $(\rho_s^n)_{0 \leq s \leq 1}$ is a constant speed displacement geodesic from $\nu_n$ to the optimal value $\mu$ with velocity vectors $(V_s^n)_{0 \leq s \leq 1}$. If $\gamma \leq 1/L$, where $L$ is the Lispchitz constant of $\nabla k$ in Assumption (A), then:*

$$2\gamma(\mathcal{F}(\nu_{n+1}) - \mathcal{F}(\mu)) \leq W_2^2(\nu_n, \mu) - W_2^2(\nu_{n+1}, \mu) - 2\gamma K(\rho^n). \tag{47}$$

*Proof.* Let $\Pi^n$ be the optimal coupling between $\nu_n$ and $\mu$, then the optimal transport between $\nu_n$ and $\mu$ is given by:

$$W_2^2(\mu, \nu_n) = \int \|X - Y\|^2 d\Pi^n(\nu_n, \mu) \tag{48}$$

Moreover, consider $Z = X - \gamma \nabla f_{\mu,\nu_n}(X)$ where $(X, Y)$ are samples from $\pi^n$. It is easy to see that $(Z, Y)$ is a coupling between $\nu_{n+1}$ and $\mu$, therefore, by definition of the optimal transport map between $\nu_{n+1}$ and $\mu$ it follows that:

$$W_2^2(\nu_{n+1}, \mu) \leq \int \|X - \gamma \nabla f_{\mu,\nu_n}(X) - Y\|^2 d\pi^n(\nu_n, \mu) \tag{49}$$

By expanding the r.h.s in (49), the following inequality holds:

$$W_2^2(\nu_{n+1}, \mu) \leq W_2^2(\nu_n, \mu) - 2\gamma \int \langle \nabla f_{\mu,\nu_n}(X), X - Y \rangle d\pi^n(\nu_n, \mu) + \gamma^2 D(\nu_n) \tag{50}$$

where $D(\nu_n) = \int \|\nabla f_{\mu,\nu_n}(X)\|^2 d\nu_n$. By Lemma 15 it holds that:

$$-2\gamma \int \nabla f_{\mu,\nu_n}(X).(X - Y) d\pi(\nu, \mu) \leq -2\gamma\left(\mathcal{F}(\nu_n) - \mathcal{F}(\mu) + K(\rho^n)\right) \tag{51}$$

where $(\rho_t^n)_{0 \leq t \leq 1}$ is a constant-speed geodesic from $\nu_n$ to $\mu$ and $K(\rho^n) := \int_0^1 \Lambda(\rho_s^n, v_s^n)(1-s)ds$. Note that when $K(\rho^n) \leq 0$ it falls back to the convex setting. Therefore, the following inequality holds:

$$W_2^2(\nu_{n+1}, \mu) \leq W_2^2(\nu_n, \mu) - 2\gamma\left(\mathcal{F}(\nu_n) - \mathcal{F}(\mu) + K(\rho^n)\right) + \gamma^2 D(\nu_n) \tag{52}$$

Now we introduce a term involving $\mathcal{F}(\nu_{n+1})$. The above inequality becomes:

$$W_2^2(\nu_{n+1}, \mu) \leq W_2^2(\nu_n, \mu) - 2\gamma\left(\mathcal{F}(\nu_{n+1}) - \mathcal{F}(\mu) + K(\rho^n)\right) \tag{53}$$
$$+ \gamma^2 D(\nu_n) - 2\gamma(\mathcal{F}(\nu_n) - \mathcal{F}(\nu_{n+1})) \tag{54}$$

It is possible to upper-bound the last two terms on the r.h.s. by a negative quantity when the step-size is small enough. This is mainly a consequence of the smoothness of the functional $\mathcal{F}$ and the fact that $\nu_{n+1}$ is obtained by following the steepest direction of $\mathcal{F}$ starting from $\nu_n$. Proposition 4 makes this statement more precise and enables to get the following inequality:

$$\gamma^2 D(\nu_n) - 2\gamma(\mathcal{F}(\nu_n) - \mathcal{F}(\nu_{n+1})) \leq -\gamma^2(1 - 3\gamma L)D(\nu_n), \tag{55}$$

where $L$ is the Lispchitz constant of $\nabla k$. Combining (54) and (55) we finally get:

$$2\gamma(\mathcal{F}(\nu_{n+1}) - \mathcal{F}(\mu)) + \gamma^2(1 - 3\gamma L)D(\nu_n) \leq W_2^2(\nu_n, \mu) - W_2^2(\nu_{n+1}, \mu) - 2\gamma K(\rho^n). \tag{56}$$

and under the condition $\gamma \leq 1/(3L)$ we recover the desired result. $\qquad\square$

We can now give the proof of the Theorem 6.

*Proof of Theorem 6.* Consider the Lyapunov function $L_j = j\gamma(\mathcal{F}(\nu_j) - \mathcal{F}(\mu)) + \frac{1}{2}W_2^2(\nu_j, \mu)$ for any iteration $j$. At iteration $j + 1$, we have:

$$L_{j+1} = j\gamma(\mathcal{F}(\nu_{j+1}) - \mathcal{F}(\mu)) + \gamma(\mathcal{F}(\nu_{j+1}) - \mathcal{F}(\mu)) + \frac{1}{2}W_2^2(\nu_{j+1}, \mu)$$

$$\leq j\gamma(\mathcal{F}(\nu_{j+1}) - \mathcal{F}(\mu)) + \frac{1}{2}W_2^2(\nu_j, \mu) - \gamma K(\rho^j)$$

$$\leq j\gamma(\mathcal{F}(\nu_j) - \mathcal{F}(\mu)) + \frac{1}{2}W_2^2(\nu_j, \mu) - \gamma K(\rho^j) - j\gamma^2(1 - \frac{3}{2}\gamma L)\int \|\nabla f_{\mu,\nu_j}(X)\|^2 d\nu_j$$

$$\leq L_j - \gamma K(\rho^j).$$

where we used Proposition 16 and Proposition 4 successively for the two first inequalities. We thus get by telescopic summation:

$$L_n \leq L_0 - \gamma \sum_{j=0}^{n-1} K(\rho^j) \tag{57}$$

Let us denote $\bar{K}$ the average value of $(K(\rho^j))_{0 \leq j \leq n}$ over iterations up to $n$. We can now write the final result:

$$\mathcal{F}(\nu_n) - \mathcal{F}(\mu) \leq \frac{W_2^2(\nu_0, \mu)}{2\gamma n} - \bar{K} \tag{58}$$

$\square$

### D.4 Lojasiewicz type inequalities

Given a probability distribution $\nu$, the *weighted Sobolev semi-norm* is defined for all squared integrable functions $f$ in $L_2(\nu)$ as $\|f\|_{\dot{H}(\nu)} = \left(\int \|\nabla f(x)\|^2 \, d\nu(x)\right)^{\frac{1}{2}}$ with the convention $\|f\|_{\dot{H}(\nu)} = +\infty$ if $f$ does not have a square integrable gradient. The *Negative weighted Sobolev distance* $\|.\|_{\dot{H}^{-1}(\nu)}$ is then defined on distributions as the dual norm of $\|.\|_{\dot{H}(\nu)}$. For convenience, we recall the definition of $\|.\|_{\dot{H}^{-1}(\nu)}$:

**Definition 5.** *Let $\nu \in \mathcal{P}_2(\mathbf{x})$, with its corresponding weighted Sobolev semi-norm $\|.\|_{\dot{H}(\nu)}$. The weighted negative Sobolev distance $\|p - q\|_{\dot{H}^{-1}(\nu)}$ between any $p$ and $q$ in $\mathcal{P}_2(\mathbf{x})$ is defined as*

$$\|p - q\|_{\dot{H}^{-1}(\nu)} = \sup_{f \in L_2(\nu), \|f\|_{\dot{H}(\nu)} \leq 1} \left| \int f(x) \, dp(x) - \int f(x) \, dq(x) \right| \tag{59}$$

*with possibly infinite values.*

There are several possible choices for the set of test functions $f$. While it is often required that $f$ vanishes at the boundary (see [41]), we do not make such restriction and rather use the definition from [45]. We refer to [50] for more discussion on the relationship between different choices for the set of test functions.

We provide now a proof for Proposition 7.

*Proof of Proposition 7.* This proof follows simply from the definition of the negative Sobolev distance. Under Assumption (A), the kernel has at most quadratic growth hence, for any $\mu, \nu \in \mathcal{P}_2(\mathcal{X})^2$, $f_{\mu,\nu} \in L_2(\nu)$. Consider $g = \|f_{\mu,\nu_t}\|_{\dot{H}(\nu_t)}^{-1} f_{\mu,\nu_t}$, then $g \in L_2(\nu_t)$ and $\|g\|_{\dot{H}(\nu_t)} \leq 1$. Therefore, we directly have:

$$\left| \int g \, d\nu_t - \int g \, d\mu \right| \leq \|\nu_t - \mu\|_{\dot{H}^{-1}(\nu_t)} \tag{60}$$

Now, recall the definition of $g$, which implies that

$$\left| \int g \, d\nu_t - \int g \, d\mu \right| = \|\nabla f_{\mu,\nu_t}\|_{L_2(\nu_t)}^{-1} \left| \int f_{\mu,\nu_t} \, d\nu_t - \int f_{\mu,\nu_t} \, d\mu \right|. \tag{61}$$

Moreover, we have that $\int f_{\mu,\nu_t}\, d\nu_t - \int f_{\mu,\nu_t}\, d\mu = \|f_{\mu,\nu_t}\|_{\mathcal{H}}^2$, since $f_{\mu,\nu_t}$ is the unnormalised witness function between $\nu_t$ and $\mu$. Combining (60) and (61) we thus get the desired Lojasiewicz inequality on $f_{\mu,\nu_t}$:

$$\|f_{\mu,\nu_t}\|_{\mathcal{H}}^2 \le \|f_{\mu,\nu_t}\|_{\dot{H}(\nu_t)}\|\mu - \nu_t\|_{\dot{H}^{-1}(\nu_t)} \tag{62}$$

where $\|f_{\mu,\nu_t}\|_{\dot{H}(\nu_t)} = \|\nabla f_{\mu,\nu_t}\|_{L_2(\nu_t)}$ by definition. Then, using Proposition 2 and recalling by assumption that: $\|\mu - \nu_t\|_{\dot{H}^{-1}(\nu_t)}^2 \le C$, we have:

$$\dot{\mathcal{F}}(\nu_t) = -\|\nabla f_{\mu,\nu_t}\|_{L_2(\nu_t)}^2 \le -\frac{1}{C}\|f_{\mu,\nu_t}\|_{\mathcal{H}}^4 = -\frac{4}{C}\mathcal{F}(\nu_t)^2 \tag{63}$$

It is clear that if $\mathcal{F}(\nu_0) > 0$ then $\mathcal{F}(\nu_t) > 0$ at all times by uniqueness of the solution. Hence, one can divide by $\mathcal{F}(\nu_t)^2$ and integrate the inequality from $0$ to some time $t$. The desired inequality is obtained by simple calculations.

Then, using Proposition 4 and (63) where $\nu_t$ is replaced by $\nu_n$ it follows:

$$\mathcal{F}(\nu_{n+1}) - \mathcal{F}(\nu_n) \le -\gamma\left(1 - \frac{3}{2}L\gamma\right)\|\nabla f_{\mu,\nu_n}\|_{L_2(\nu_n)}^2 \le -\frac{4}{C}\gamma\left(1 - \frac{3}{2}\gamma L\right)\mathcal{F}(\nu_n)^2.$$

Dividing by both sides of the inequality by $\mathcal{F}(\nu_n)\mathcal{F}(\nu_{n+1})$ and recalling that $\mathcal{F}(\nu_{n+1}) \le \mathcal{F}(\nu_n)$ it follows directly that:

$$\frac{1}{\mathcal{F}(\nu_n)} - \frac{1}{\mathcal{F}(\nu_{n+1})} \le -\frac{4}{C}\gamma\left(1 - \frac{3}{2}\gamma L\right).$$

The proof is concluded by summing over $n$ and rearranging the terms. $\qquad\square$

## D.5 A simple example

Consider a gaussian target distribution $\mu(x) = \mathcal{N}(a, \Sigma)$ and initial distribution $\nu_0 = \mathcal{N}(a_0, \Sigma_0)$. In this case it is sufficient to use a kernel that captures the first and second moments of the distribution. We simply consider a kernel of the form $k(x, y) = (x^\top y)^2 + x^\top y$. In this case, it is easy to see by simple computations that the following equation holds:

$$\dot{X}_t = -(\Sigma_t - \Sigma + a_t a_t^\top - aa^\top)X_t - (a_t - a), \qquad \forall t \ge 0 \tag{64}$$

Where $a_t$ and $\Sigma_t$ are the mean and covariance matrix of $\nu_t$ and satisfy the equations:

$$\dot{\Sigma}_t = -(S_t\Sigma_t + \Sigma_t S_t) \tag{65}$$
$$\dot{a}_t = -S_t a_t - (a_t - a). \tag{66}$$

Where we introduced $S_t = \Sigma_t - \Sigma + a_t a_t^\top - aa^\top$ for simplicity. (64) implies that $\nu_t$ is in fact a gaussian distribution since $X_t$ is obtained by summing gaussian increments. The same conclusion can be reached by solving the corresponding continuity equation. Thus we will be only interested in the behavior of $a_t$ and $\Sigma_t$. First we can express the squared MMD in terms of those parameters:

$$MMD^2(\mu, \nu_t) = \|S_t\|^2 + \|a_t - a\|^2. \tag{67}$$

Since $a_t$ and $\Sigma_t$ are obtained from the gradient flow of the MMD, it follows that $\|a_t - a\|^2$ and $\|S_t\|^2$ remain bounded. Moreover, the Negative Sobolev distance is obtained by solving a finite dimensional quadratic problem and can be simply written as:

$$D(\mu, \nu_t) = tr(Q_t\Sigma_t Q_t) + \|a_t - a\|^2 \tag{68}$$

where $Q_t$ is the unique solution of the Lyapounov equation:

$$\Sigma_t Q_t + Q_t\Sigma_t = \Sigma_t - \Sigma + (a_t - a)(a_t - a)^\top := G_t. \tag{69}$$

We first consider the one dimensional case, for which (69) has a particularly simple solution and allows to provide a closed form expression for the negative Sobolev distance:

$$Q_t = \frac{G_t}{2\Sigma_t}, \qquad D(\mu, \nu_t) = \frac{G_t^2}{4\Sigma_t} + (a_t - a)^2. \tag{70}$$

Recalling (67) and that $MMD^2(\mu, \nu_t)$ is bounded at all times by definition of $\nu_t$, it follows that both $G_t$ and $a_t - a$ are also bounded. Hence, it is easy to see that $D(\mu, \nu_t)$ will remain bounded iff $\Sigma_t$

remains bounded away from 0. This analysis generalizes the higher dimensions using [4, Lemma 3.2 (iii)] which provides an expression for $Q_t$ in terms of $G_t$ and the singular value decomposition of $\Sigma_t = U_t D_t U_t^\top$:

$$Q_t = U_t \left( \left( \frac{1}{(D_t)_i + (D_t)_j} \right) \odot U_t^\top G_t U_t \right) U_t^\top. \tag{71}$$

Here, $\odot$ denotes the Hadamard product of matrices. It is easy to see from this expression that $D(\mu, \nu_t)$ will be bounded if all singular values $((D_t)_i)_{1 \le i \le d}$ of $\Sigma_t$ remain bounded away from 0.

## D.6 Lojasiewicz-type inequalities for $\mathcal{F}$ under different metrics

The Wasserstein gradient flow of $\mathcal{F}$ can be seen as the continuous-time limit of the so called minimizing movement scheme [1]. Such proximal scheme is defined using an initial distribution $\nu_0$, a step-size $\tau$, and an iterative update equation:

$$\nu_{n+1} \in \arg \min_{\nu} \mathcal{F}(\nu) + \frac{1}{2\tau} W_2^2(\nu, \nu_n). \tag{72}$$

In [1], it is shown that the continuity equation $\partial_t \nu_t = div(\nu_t \nabla f_{\mu,\nu_t})$ can be obtained as the limit when $\tau \to 0$ of (72) using suitable interpolations between the elements $\nu_n$. In [47], a different transport equation that includes a birth-death term is considered:

$$\partial_t \nu_t = \beta div(\nu_t \nabla f_{\mu,\nu_t}) + \alpha(f_{\mu,\nu_t} - \int f_{\mu,\nu_t}(x) \, \mathrm{d}\nu_t(x))\nu_t \tag{73}$$

When $\beta = 0$ and $\alpha = 1$, it is shown formally in [47] that the above dynamics corresponds to the limit of a proximal scheme using the KL instead of the Wasserstein distance. For general $\beta$ and $\alpha$, (74) corresponds to the limit of a different proximal scheme where $W_2^2(\nu, \nu_n)$ is replaced by the Wasserstein-Fisher-Rao distance $d_{\alpha,\beta}^2(\nu, \nu_n)$ (see [16, 33, 37]). $d_{\alpha,\beta}^2(\nu, \nu_n)$ is an interpolation between the squared Wasserstein distance ($\beta = 1$ and $\alpha = 0$) and the squared Fisher-Rao distance as defined in [16, Definition 6] ($\beta = 0$ and $\alpha = 1$). Such scheme is consistent with the one proposed in [47] and which uses the $KL$. In fact, as we will show later, both the $KL$ and the Fisher-Rao distance have the same local behavior therefore both proximal schemes are expected to be equivalent in the limit when $\tau \to 0$.

Under (74), the time evolution of $\mathcal{F}$ is given by [47, Proposition 3.1]:

$$\dot{\mathcal{F}}(\nu_t) = -\beta \int \|\nabla f_{\mu,\nu_t}\|^2 \, \mathrm{d}\nu_t(x) - \alpha \int \left| f_{\mu,\nu_t}(x) - \int f_{\mu,\nu_t}(x') \, \mathrm{d}\nu_t(x') \right|^2 \, \mathrm{d}\nu_t(x) \tag{74}$$

We would like to apply the same approach as in Section 3.2 to provide a condition on the convergence of (74). Hence we first introduce an analogue to the Negative Sobolev distance in Definition 1 by duality:

$$D_\nu(p, q) = \sup_{\substack{g \in L_2(\nu) \\ \beta\|\nabla g\|_{L_2(\nu)}^2 + \alpha\|g - \bar{g}\|_{L_2(\nu)}^2 \le 1}} \left| \int g(x) \, \mathrm{d}p(x) - \int g(x) \, \mathrm{d}q(x) \right| \tag{75}$$

where $\bar{g}$ is simply the expectation of $g$ under $\nu$. Such quantity defines a distance, since it is the dual of a semi-norm. Now using the particular structure of the MMD, we recall that $f_{\mu,\nu} \in L_2(\nu)$ and that $\beta\|\nabla f\|_{L_2(\nu)}^2 + \alpha\|f - \bar{f}\|_{L_2(\nu)}^2 < \infty$. Hence for a particular $g$ of the form:

$$g = \frac{f_{\mu,\nu}}{\left( \beta\|\nabla f_{\mu,\nu}\|_{L_2(\nu)}^2 + \alpha\|f_{\mu,\nu} - \bar{f}_{\mu,\nu}\|_{L_2(\nu)}^2 \right)^{\frac{1}{2}}}$$

the following inequality holds:

$$D_\nu(\mu, \nu) \ge \frac{\left| \int f_{\mu,\nu} \, \mathrm{d}\nu(x) - \int f_{\mu,\nu} \, \mathrm{d}\mu(x) \right|}{\left( \beta\|\nabla f_{\mu,\nu}\|_{L_2(\nu)}^2 + \alpha\|f_{\mu,\nu} - \bar{f}_{\mu,\nu}\|_{L_2(\nu)}^2 \right)^{\frac{1}{2}}}.$$

But since $f_{\mu,\nu}$ is the unnormalised witness function between $\mu$ and $\nu$ we have that $2\mathcal{F}(\nu) = \left| \int f_{\mu,\nu} \, \mathrm{d}\nu(x) - \int f_{\mu,\nu} \, \mathrm{d}\mu(x) \right|$. Hence one can write that:

$$D_\nu^2(\mu,\nu) \left( \beta \|\nabla f_{\mu,\nu}\|_{L_2(\nu)}^2 + \alpha \|f_{\mu,\nu} - \bar{f}_{\mu,\nu}\|_{L_2(\nu)}^2 \right) \geq 4\mathcal{F}^2(\nu) \tag{76}$$

Now provided that $D_\nu^2(\mu,\nu_t)$ remains bounded at all time $t$ by some constant $C > 0$ one can easily deduce a rate of convergence for $\mathcal{F}(\nu_t)$ just as in Proposition 7. In fact, in the case when $\beta = 1$ and $\alpha = 0$ one recovers Proposition 7. Another interesting case is when $\beta = 0$ and $\alpha = 1$. In this case, $D_\nu(p,q)$ is defined for $p$ and $q$ such that the difference $p - q$ is absolutely continuous w.r.t. $\nu$. Moreover, $D_\nu(p,q)$ has the simple expression:

$$D_\nu(p,q) = \int \left( \frac{p-q}{\nu}(x) \right)^2 \mathrm{d}\nu(x)$$

where $\frac{p-q}{\nu}$ denotes the radon nikodym density of $p - q$ w.r.t. $\nu$. More importantly, $D_\nu^2(\mu,\nu)$ is exactly equal to $\chi^2(\mu\|\nu)^{\frac{1}{2}}$. As we will show now, $(\chi^2)^{\frac{1}{2}}$ turns out to be a linearization of $\sqrt{2}KL^{\frac{1}{2}}$ and the Fisher-Rao distance.

**Linearization of the KL and the Fisher-Rao distance.** We first show the result for the KL. Given a probability distribution $\nu'$ that is absolutely continuous w.r.t to $\nu$ and for $0 < \epsilon < 1$ denote by $G(\epsilon) := KL(\nu\|(\nu + \epsilon(\nu' - \nu))$. It can be shown that $G(\epsilon) = \frac{1}{2}\chi^2(\nu'\|\nu)\epsilon^2 + o(\epsilon^2)$. To see this, one needs to perform a second order Taylor expansion of $G(\epsilon)$ at $\epsilon = 0$. Exchanging the derivatives and the integral, $\dot{G}(\epsilon)$ and $\ddot{G}(\epsilon)$ are both given by:

$$\dot{G}(\epsilon) = -\int \frac{\mu - \nu}{\nu + \epsilon(\mu - \nu)} \, \mathrm{d}\nu$$

$$\ddot{G}(\epsilon) = \int \frac{(\nu - \mu)^2}{(\nu + \epsilon(\mu - \nu))^2} \, \mathrm{d}\nu$$

Hence, we have for $\epsilon = 0$: $\dot{G}(0) = 0$ and $\ddot{G}(0) = \chi^2(\mu\|\nu)$. Therefore, it follows: $G(\epsilon) = \frac{1}{2}\chi^2(\mu\|\nu)\epsilon^2 + o(\epsilon^2)$, which means that

$$\lim_{\epsilon \to 0} \frac{1}{\epsilon} \left[ 2KL\left(\nu\|\nu + \epsilon(\nu' - \nu)\right) \right]^{\frac{1}{2}} = \chi^2(\nu'\|\nu)^{\frac{1}{2}}.$$

The same approach can be used for the Fisher-Rao distance $d_{0,1}(\nu, \nu')$. From [16, Theorem 3.1] we have that:

$$d_{0,1}^2(\nu, \nu') = 2 \int (\sqrt{\nu(x)} - \sqrt{\nu'(x)})^2 \, \mathrm{d}x$$

where $\nu$ and $\nu'$ are assumed to have a density w.r.t. Lebesgue measure. Using the exact same approach as for the KL one easily show that $\lim_{\epsilon \to 0} \frac{1}{\epsilon} \left[ 2d_{0,1}^2\left(\nu\|\nu + \epsilon(\nu' - \nu)\right) \right]^{\frac{1}{2}} = \chi^2(\nu'\|\nu)^{\frac{1}{2}}$.

**Linearization of the $W_2$.** Similarly, it can be shown that the *Negative weighted Sobolev distance* is a linearization of the $W_2$ under suitable conditions. We recall here [59, Theorem 7.26] which relates the two quantities:

**Theorem 17.** *Let $\nu \in \mathcal{P}(\mathcal{X})$ be a probability measure with finite second moment, absolutely continuous w.r.t the Lebesgue measure and let $h \in L^\infty(\mathcal{X})$ with $\int h(x) \, \mathrm{d}\nu(x) = 0$. Then*

$$\|h\|_{\dot{H}^{-1}(\nu)} \leq \liminf_{\epsilon \to 0} \frac{1}{\epsilon} W_2(\nu, (1 + \epsilon h)\nu).$$

Theorem 17 implies that for any probability distribution $\nu'$ that has a bounded density w.r.t. to $\nu$ one has:

$$\|\nu' - \nu\|_{\dot{H}^{-1}(\nu)} \leq \liminf_{\epsilon \to 0} \frac{1}{\epsilon} W_2(\nu, \nu + \epsilon(\nu' - \nu)).$$

To get the converse inequality, one needs to assume that the support of $\nu$ is $\mathcal{X}$. Proposition 18 provides such inequality and uses techniques from [45].

**Proposition 18.** *Let $\nu \in \mathcal{P}(\mathcal{X})$ be a probability measure with finite second moment, absolutely continuous w.r.t the Lebesgue measure with support equal to $\mathcal{X}$ and let $h \in L^\infty(\mathcal{X})$ with $\int h(x)\,d\nu(x) = 0$ and $1 + h \geq 0$. Then*

$$\limsup_{\epsilon \to 0} \frac{1}{\epsilon} W_2(\nu, (1+\epsilon h)\nu) \leq \|h\|_{\dot{H}^{-1}(\nu)}$$

*Proof.* Consider the elliptic equation: $\nu h + div(\nu \nabla F) = 0$ with Neumann boundary condition on $\partial \mathcal{X}$. Such equation admits a unique solution $F$ in $\dot{H}(\nu)$ up to a constant since $\nu$ is supported on all of $\mathcal{X}$ (see [44, Section 7 (Linearizations)]). Moreover, we have that $\int F(x)h(x)\,d\nu(x) = \int \|\nabla F(x)\|^2\,d\nu(x)$ which implies that $\|h\|_{\dot{H}^{-1}(\nu)} \geq \|F\|_{\dot{H}(\nu)}$. Now consider the path: $s_u = (1 + u\epsilon h)\nu$ for $u \in [0,1]$. $s_u$ is a probability distribution for all $u \in [0,1]$ with $s_0 = \nu$ and $s_1 = (1+\epsilon h)\nu$. It is easy to see that $s_u$ satisfies the continuity equation:

$$\partial_u s_u + div(s_u V_u) = 0$$

with $V_u = \frac{\epsilon \nabla F}{1 + u\epsilon h}$. Indeed, for any smooth test function $f$ one has:

$$\frac{d}{du}\int f(x)\,ds_u(x) = \epsilon \int f(x)h(x)\,d\nu(x) = \epsilon \int \nabla f(x).\nabla F(x)\,d\nu(x) = \int \nabla f(x).V_u(x)\,ds_u(x).$$

We used the definition of $F$ for the second equality and that $\nu$ admits a density w.r.t. to $s_u$ provided that $\epsilon$ is small enough. Such density is given by $1/(1 + u\epsilon h)$ and is positive and bounded when $\epsilon \leq \frac{1}{2\|h\|_\infty}$. Now, using the Benamou-Brenier formula for $W_2(\nu, (1+\epsilon h)\nu)$ one has in particular that:

$$W_2(\nu, (1+\epsilon h)\nu) \leq \int \|V_u\|_{L^2(s_u)}\,du$$

Using the expressions of $V_u$ and $s_u$, one gets by simple computation:

$$W_2(\nu, (1+\epsilon h)\nu) \leq \epsilon \int \left( \int \frac{\|\nabla F(x)\|^2}{1 - u\epsilon + u\epsilon(h+1)}\,d\nu(x) \right)^{\frac{1}{2}} du$$

$$\leq \epsilon \left( \int \|\nabla F(x)\|^2\,d\nu(x) \right)^{\frac{1}{2}} \int_0^1 (1-u\epsilon)^{-\frac{1}{2}}\,du.$$

Finally, $\epsilon \int_0^1 (1-u\epsilon)^{-\frac{1}{2}}\,du = 2(1 - \sqrt{1-\epsilon}) \to 1$ when $\epsilon \to 0$, hence:

$$\limsup_{\epsilon \to 0} W_2(\nu, (1+\epsilon h)) \leq \|F\|_{\dot{H}(\nu)} \leq \|h\|_{\dot{H}^{-1}(\nu)}.$$

$\square$

Theorem 17 and Proposition 18 allow to conclude that $\lim_{\epsilon \to 0} \frac{1}{\epsilon} W_2(\nu, \nu + \epsilon(\nu' - \nu)) = \|\nu - \nu'\|_{\dot{H}^{-1}(\nu)}$ for any $\nu'$ that has a bounded density w.r.t. $\nu$.

By analogy, one could wonder if $D$ is also a linearization of the the Wasserstein-Fisher-Rao distance. We leave such question for future work.

# E  Algorithms

## E.1  Noisy Gradient flow of the MMD

*Proof of Proposition 8.* To simplify notations, we write $\mathcal{D}_{\beta_n}(\nu_n) = \int \|V(x + \beta_n u)\|^2 g(u)\,d\nu_n\,du$ where $V := \nabla f_{\mu,\nu_n}$ and $g$ is the density of a standard gaussian. The symbol $\otimes$ denotes the product of two independent probability distributions. Recall that a sample $x_{n+1}$ from $\nu_{n+1}$ is obtained using $x_{n+1} = x_n - \gamma V(x_n + \beta_n u_n)$ where $x_n$ is a sample from $\nu_n$ and $u_n$ is a sample from a standard gaussian distribution that is independent from $x_n$. Moreover, by assumption $\beta_n$ is a non-negative scalar satisfying:

$$8\lambda^2 \beta_n^2 \mathcal{F}(\nu_n) \leq \mathcal{D}_{\beta_n}(\nu_n) \tag{77}$$

Consider now the map $(x, u) \mapsto s_t(x) = x - \gamma t V(x + \beta_n u)$ for $0 \leq t \leq 1$, then $\nu_{n+1}$ is obtained as a push-forward of $\nu_n \otimes g$ by $s_1$: $\nu_{n+1} = (s_1)_\#(\nu_n \otimes g)$. Moreover, the curve $\rho_t = (s_t)_\#(\nu_n \otimes g)$

is a path from $\nu_n$ to $\nu_{n+1}$. We know by Proposition 21 that $\nabla f_{\mu,\nu_n}$ is $2L$-Lipschitz, thus using $\phi(x, u) = -\gamma V(x + \beta_n u)$, $\psi(x, u) = x$ and $q = \nu_n \otimes g$ in Lemma 22 it follows that $\mathcal{F}(\rho_t)$ is differentiable in $t$ with:

$$\dot{\mathcal{F}}(\rho_t) = \int \nabla f_{\mu,\rho_t}(s_t(x)).(-\gamma V(x + \beta_n u))g(u)\,\mathrm{d}\nu_n(x)\,\mathrm{d}u$$

Moreover, $\dot{\mathcal{F}}(\rho_0)$ is given by $\dot{\mathcal{F}}(\rho_0) = -\gamma \int V(x).V(x + \beta_n u)g(u)\,\mathrm{d}\nu_n(x)\,\mathrm{d}u$ and the following estimate holds:

$$|\dot{\mathcal{F}}(\rho_t) - \dot{\mathcal{F}}(\rho_0)| \leq 3\gamma^2 Lt \int \|V(x + \beta_n u)\|^2 g(u)\,\mathrm{d}\nu_n(x)\,\mathrm{d}u = 3\gamma^2 Lt \mathcal{D}_{\beta_n}(\nu_n). \qquad (78)$$

Using the absolute continuity of $\mathcal{F}(\rho_t)$, one has $\mathcal{F}(\nu_{n+1}) - \mathcal{F}(\nu_n) = \dot{\mathcal{F}}(\rho_0) + \int_0^1 \dot{\mathcal{F}}(\rho_t) - \dot{\mathcal{F}}(\rho_0)\,\mathrm{d}t$. Combining with (78) and using the expression of $\dot{\mathcal{F}}(\rho_0)$, it follows that:

$$\mathcal{F}(\nu_{n+1}) - \mathcal{F}(\nu_n) \leq -\gamma \int V(x).V(x + \beta_n u)g(u)\,\mathrm{d}\nu_n(x)\,\mathrm{d}u + \frac{3}{2}\gamma^2 L \mathcal{D}_{\beta_n}(\nu_n). \qquad (79)$$

Adding and subtracting $\gamma \mathcal{D}_{\beta_n}(\nu_n)$ in (79) it follows directly that:

$$\mathcal{F}(\nu_{n+1}) - \mathcal{F}(\nu_n) \leq -\gamma(1 - \frac{3}{2}\gamma L)\mathcal{D}_{\beta_n}(\nu_n)$$
$$+ \gamma \int (V(x + \beta_n u) - V(x)).V(x + \beta_n u)g(u)\,\mathrm{d}\nu_n(x)\,\mathrm{d}u \qquad (80)$$

We shall control now the last term in (80). Recall now that for all $1 \leq i \leq d$, $V_i(x) = \partial_i f_{\mu,\nu_n}(x) = \langle f_{\mu,\nu_n}, \partial_i k(x, .)\rangle$ where we used the reproducing property for the derivatives of $f_{\mu,\nu_n}$ in $\mathcal{H}$ (see Appendix A.1). Therefore, it follows by Cauchy-Schwartz in $\mathcal{H}$ and using Assumption (D):

$$\|V(x + \beta_n u) - V(x)\|^2 \leq \|f_{\mu,\nu_n}\|_{\mathcal{H}}^2 \left(\sum_{i=1}^d \|\partial_i k(x + \beta_n u, .) - \partial_i k(x, .)\|_{\mathcal{H}}^2\right)$$
$$\leq \lambda^2 \beta_n^2 \|f_{\mu,\nu_n}\|_{\mathcal{H}}^2 \|u\|^2$$

for all $x, u \in \mathcal{X}$. Now integrating both sides w.r.t. $\nu_n$ and $g$ and recalling that $g$ is a standard gaussian, we have:

$$\int \|V(x + \beta_n u) - V(x)\|^2 g(u)\,\mathrm{d}\nu_n(x)\,\mathrm{d}u \leq \lambda^2 \beta_n^2 \|f_{\mu,\nu_n}\|_{\mathcal{H}}^2 \qquad (81)$$

Getting back to (80) and applying Cauchy-Schwarz in $L_2(\nu_n \otimes g)$ it follows:

$$\mathcal{F}(\nu_{n+1}) - \mathcal{F}(\nu_n) \leq -\gamma(1 - \frac{3}{2}\gamma L)\mathcal{D}_{\beta_n}(\nu_n) + \gamma \lambda \beta_n \|f_{\mu,\nu_n}\|_{\mathcal{H}} \mathcal{D}_{\beta_n}^{\frac{1}{2}}(\nu_n) \qquad (82)$$

It remains to notice that $\|f_{\mu,\nu_n}\|_{\mathcal{H}}^2 = 2\mathcal{F}(\nu_n)$ and that $\beta_n$ satisfies (77) to get:

$$\mathcal{F}(\nu_{n+1}) - \mathcal{F}(\nu_n) \leq -\frac{\gamma}{2}(1 - \frac{3}{2}\gamma L)\mathcal{D}_{\beta_n}(\nu_n).$$

We introduce now $\Gamma = 4\gamma(1 - \frac{3}{2}\gamma L)\lambda^2$ to simplify notation and prove the second inequality. Using (77) again in the above inequality we directly have: $\mathcal{F}(\nu_{n+1}) - \mathcal{F}(\nu_n) \leq -\Gamma \beta_n^2 \mathcal{F}(\nu_n)$. One can already deduce that $\Gamma \beta_n^2$ is necessarily smaller than 1. Hence, taking $\mathcal{F}(\nu_n)$ to the r.h. side and iterating over $n$ it follows that:

$$\mathcal{F}(\nu_n) \leq \mathcal{F}(\nu_0) \prod_{i=0}^{n-1}(1 - \Gamma \beta_n^2)$$

Simply using that $1 - \Gamma \beta_n^2 \leq e^{-\Gamma \beta_n^2}$ leads to the desired upper-bound $\mathcal{F}(\nu_n) \leq \mathcal{F}(\nu_0)e^{-\Gamma \sum_{i=0}^{n-1} \beta_n^2}$.

$\square$

## E.2  Sample-based approximate scheme

*Proof of Theorem 9.* Let $(u_n^i)_{1 \le i \le N}$ be i.i.d standard gaussian variables and $(x_0^i)_{1 \le i \le N}$ i.i.d. samples from $\nu_0$. We consider $(x_n^i)_{1 \le i \le N}$ the particles obtained using the approximate scheme (21): $x_{n+1}^i = x_n^i - \gamma \nabla f_{\hat{\mu}, \hat{\nu}_n}(x_n^i + \beta_n u_n^i)$ starting from $(x_0^i)_{1 \le i \le N}$, where $\hat{\nu}_n$ is the empirical distribution of these $N$ interacting particles. Similarly, we denote by $(\bar{x}_n^i)_{1 \le i \le N}$ the particles obtained using the exact update equation (17): $\bar{x}_{n+1}^i = \bar{x}_n^i - \gamma \nabla f_{\mu, \nu_n}(\bar{x}_n^i + \beta_n u_n^i)$ also starting from $(x_0^i)_{1 \le i \le N}$. By definition of $\nu_n$ we have that $(\bar{x}_n^i)_{1 \le i \le N}$ are i.i.d. samples drawn from $\nu_n$ with empirical distribution denoted by $\bar{\nu}_n$. We will control the expected error $c_n$ defined as $c_n^2 = \frac{1}{N} \sum_{i=1}^N \mathbb{E}\left[\|x_n^i - \bar{x}_n^i\|^2\right]$. By recursion, we have:

$$
c_{n+1} = \frac{1}{\sqrt{N}} \left( \sum_{i=1}^N \mathbb{E}\left[ \left\| x_n^i - \bar{x}_n^i - \gamma \left( \nabla f_{\hat{\mu}, \hat{\nu}_n}(x_n^i + \beta_n u_n^i) - \nabla f_{\mu, \nu_n}(\bar{x}_n^i + \beta_n u_n^i) \right) \right\|^2 \right] \right)^{\frac{1}{2}}
$$

$$
\le c_n + \frac{\gamma}{\sqrt{N}} \left[ \sum_{i=1}^N \mathcal{E}_i \right]^{\frac{1}{2}} + \frac{\gamma}{\sqrt{N}} \left[ \sum_{i=1}^N \mathcal{G}_i \right]^{\frac{1}{2}}
$$

$$
+ \frac{\gamma}{\sqrt{N}} \left( \sum_{i=1}^N \mathbb{E}\left[ \left\| \nabla f_{\mu, \hat{\nu}_n}(x_n^i + \beta_n u_n^i) - \nabla f_{\mu, \bar{\nu}_n}(\bar{x}_n^i + \beta_n u_n^i) \right\|^2 \right] \right)^{\frac{1}{2}}
$$

$$
\le c_n + 2\gamma L \left( c_n + \mathbb{E}\left[ W_2(\hat{\nu}_n, \bar{\nu}_n)^2 \right]^{\frac{1}{2}} \right) + \frac{\gamma}{\sqrt{N}} \left[ \sum_{i=1}^N \mathcal{E}_i \right]^{\frac{1}{2}} + \frac{\gamma}{\sqrt{N}} \left[ \sum_{i=1}^N \mathcal{G}_i \right]^{\frac{1}{2}}
$$

where the second line follows from a simple triangular inequality and the last line is obtained recalling that $\nabla f_{\mu,\nu}(x)$ is jointly $2L$ Lipschitz in $x$ and $\nu$ by Proposition 21. Here, $\mathcal{E}_i$ represents the error between $\bar{\nu}_n$ and $\nu_n$ while $\mathcal{G}_i$ represents the error between $\hat{\mu}$ and $\mu$ and are given by:

$$
\mathcal{E}_i = \mathbb{E}\left[ \left\| \nabla f_{\mu, \bar{\nu}_n}(\bar{x}_n^i + \beta_n u_n^i) - \nabla f_{\mu, \nu_n}(\bar{x}_n^i + \beta_n u_n^i) \right\|^2 \right]
$$

$$
\mathcal{G}_i = \mathbb{E}\left[ \left\| \nabla f_{\hat{\mu}, \hat{\nu}_n}(x_n^i + \beta_n u_n^i) - \nabla f_{\mu, \hat{\nu}_n}(x_n^i + \beta_n u_n^i) \right\|^2 \right]
$$

We will first control the error term $\mathcal{E}_i$. To simplify notations, we write $y^i = \bar{x}_n^i + \beta_n u_n^i$. Recalling the expression of $\nabla f_{\mu,\nu}$ from Proposition 21 and expanding the squared norm in $\mathcal{E}_i$, it follows:

$$
\mathcal{E}_i = \mathbb{E}\left[ \left\| \frac{1}{N} \sum_{j=1}^N \nabla k(y^i, \bar{x}_n^j) - \int \nabla k(y^i, x) d\nu_n(x) \right\|^2 \right]
$$

$$
= \frac{1}{N^2} \sum_{j=1}^N \mathbb{E}\left[ \left\| \nabla k(y^i, \bar{x}_n^j) - \int \nabla k(y^i, x) d\nu_n(x) \right\|^2 \right]
$$

$$
\le \frac{L^2}{N^2} \sum_{j=1}^N \mathbb{E}\left[ \left\| \bar{x}_n^j - \int x \, d\nu_n(x) \right\|^2 \right] = \frac{L^2}{N} var(\nu_n).
$$

The second line is obtained using the independence of the auxiliary samples $(\bar{x}_n^i)_{1 \le i \le N}$ and recalling that they are distributed according to $\nu_n$. The last line uses the fact that $\nabla k(y, x)$ is $L$-Lipschitz in $x$ by Assumption (A). To control the variance $var(\nu_n)$ we use Lemma 19 which implies that $var(\nu_n)^{\frac{1}{2}} \le (B + var(\nu_0)^{\frac{1}{2}}) e^{LT}$ for all $n \le \frac{2T}{\gamma}$. For $\mathcal{G}_i$, it is sufficient to expand again the squared norm and recall that $\nabla k(y, x)$ is $L$-Lipschitz in $x$ which then implies that $\mathcal{G}_i \le \frac{L^2}{M} var(\mu)$. Finally, one can observe that $\mathbb{E}[W_2^2(\hat{\nu}_n, \bar{\nu}_n)] \le \frac{1}{N} \sum_{i=1}^N \mathbb{E}\left[\|x_n^i - \bar{x}_n^i\|^2\right] = c_n^2$, hence $c_n$ satisfies the recursion:

$$
c_{n+1} \le (1 + 4\gamma L) c_n + \frac{\gamma L}{\sqrt{N}} (B + var(\nu_0)^{\frac{1}{2}}) e^{2LT} + \frac{\gamma L}{\sqrt{M}} var(\mu).
$$

Using Lemma 26 to solve the above inequality, it follows that:

$$
c_n \le \frac{1}{4} \left( \frac{1}{\sqrt{N}} (B + var(\nu_0)^{\frac{1}{2}}) e^{2LT} + \frac{1}{\sqrt{M}} var(\mu) \right) (e^{4LT} - 1)
$$

$\square$

**Lemma 19.** *Consider an initial distribution $\nu_0$ with finite variance, a sequence $(\beta_n)_{n\geq 0}$ of non-negative numbers bounded by $B < \infty$ and define the sequence of probability distributions $\nu_n$ of the process (17):*

$$x_{n+1} = x_n - \gamma \nabla f_{\mu,\nu_n}(x_n + \beta_n u_n) \qquad x_0 \sim \nu_0$$

*where $(u_n)_{n\geq 0}$ are standard gaussian variables. Under Assumption (A), the variance of $\nu_n$ satisfies for all $T > 0$ and $n \leq \frac{T}{\gamma}$ the following inequality:*

$$var(\nu_n)^{\frac{1}{2}} \leq (B + var(\nu_0)^{\frac{1}{2}})e^{2TL}$$

*Proof.* Let $g$ be the density of a standard gaussian. Denote by $(x, u)$ and $(x', u')$ two independent samples from $\nu_n \otimes g$. The idea is to find a recursion from $var(\nu_n)$ to $var(\nu_{n+1})$:

$$var(\nu_{n+1})^{\frac{1}{2}} = \left( \mathbb{E}\left[ \|x - \mathbb{E}\left[x'\right] - \gamma \nabla f_{\mu,\nu_n}(x + \beta_n u) + \gamma \mathbb{E}\left[\nabla f_{\mu,\nu_n}(x' + \beta_n u')\right]\|^2 \right] \right)^{\frac{1}{2}}$$

$$\leq var(\nu_n)^{\frac{1}{2}} + \gamma \left( \mathbb{E}\left[ \|\nabla f_{\mu,\nu_n}(x + \beta_n u) - \mathbb{E}\left[\nabla f_{\mu,\nu_n}(x' + \beta_n u')\right]\|^2 \right] \right)^{\frac{1}{2}}$$

$$\leq var(\nu_n)^{\frac{1}{2}} + 2\gamma L \mathbb{E}_{\substack{x,x'\sim \nu_n \\ u,u'\sim g}} \left[ \|x + \beta_n u - x' + \beta_n u'\|^2 \right]^{\frac{1}{2}}$$

$$\leq var(\nu_n)^{\frac{1}{2}} + 2\gamma L (var(\nu_n)^{\frac{1}{2}} + \beta_n)$$

The second and last lines are obtained using a triangular inequality while the third line uses that $\nabla f_{\mu,\nu_n}(x)$ is $2L$-Lipschitz in $x$ by Proposition 21. Recalling that $\beta_n$ is bounded by $B$ it is easy to conclude using Lemma 26. $\square$

# F    Connection with Neural Networks

In this sub-section we establish a formal connection between the MMD gradient flow defined in (5) and neural networks optimization. Such connection holds in the limit of infinitely many neurons and is based on the formulation in [48]. To remain consistent with the rest of the paper, the parameters of a network will be denoted by $x \in \mathcal{X}$ while the input and outputs will be denoted as $z$ and $y$. Given a neural network or any parametric function $(z, x) \mapsto \psi(z, x)$ with parameter $x \in \mathcal{X}$ and input data $z$ we consider the supervised learning problem:

$$\min_{(x_1,\ldots,x_m)\in\mathcal{X}} \frac{1}{2} \mathbb{E}_{(y,z)\sim p} \left[ \left\| y - \frac{1}{m} \sum_{i=1}^{m} \psi(z, x_i) \right\|^2 \right] \tag{83}$$

where $(y, z) \sim p$ are samples from the data distribution and the regression function is an average of $m$ different networks. The formulation in (83) includes any type of networks. Indeed, the averaged function can itself be seen as one network with augmented parameters $(x_1,\ldots,x_m)$ and any network can be written as an average of sub-networks with potentially shared weights. In the limit $m \to \infty$, the average can be seen as an expectation over the parameters under some probability distribution $\nu$. This leads to an expected network $\Psi(z, \nu) = \int \psi(z, x)\, d\nu(x)$ and the optimization problem in (83) can be lifted to an optimization problem in $\mathcal{P}_2(\mathcal{X})$ the space of probability distributions:

$$\min_{\nu\in\mathcal{P}_2(\mathcal{X})} \mathcal{L}(\nu) := \frac{1}{2} \mathbb{E}_{(y,z)\sim p} \left[ \left\| y - \int \psi(z, x)\, d\nu(x) \right\|^2 \right] \tag{84}$$

For convenience, we consider $\bar{\mathcal{L}}(\nu)$ the function obtained by subtracting the variance of $y$ from $\mathcal{L}(\nu)$, i.e.: $\bar{\mathcal{L}}(\nu) = \mathcal{L}(\nu) - var(y)$. When the model is well specified, there exists $\mu \in \mathcal{P}_2(\mathcal{X})$ such that $\mathbb{E}_{y\sim\mathbb{P}(.|z)}[y] = \int \psi(z, x)\, d\mu(x)$. In that case, the cost function $\bar{\mathcal{L}}$ matches the functional $\mathcal{F}$ defined in (3) for a particular choice of the kernel $k$. More generally, as soon as a global minimizer for (84) exists, Proposition 20 relates the two losses $\bar{\mathcal{L}}$ and $\mathcal{F}$.

**Proposition 20.** *Assuming a global minimizer of (84) is achieved by some $\mu \in \mathcal{P}_2(\mathcal{X})$, the following inequality holds for any $\nu \in \mathcal{P}_2(\mathcal{X})$:*

$$\left( \bar{\mathcal{L}}(\mu)^{\frac{1}{2}} + \mathcal{F}^{\frac{1}{2}}(\nu) \right)^2 \geq \bar{\mathcal{L}}(\nu) \geq \mathcal{F}(\nu) + \bar{\mathcal{L}}(\mu) \tag{85}$$

*where $\mathcal{F}(\nu)$ is defined by (3) with a kernel $k$ constructed from the data as an expected product of networks:*

$$k(x, x') = \mathbb{E}_{z \sim \mathbb{P}} \left[ \psi(z, x)^T \psi(z, x') \right] \tag{86}$$

*Moreover, $\bar{\mathcal{L}} = \mathcal{F}$ iif $\bar{\mathcal{L}}(\mu) = 0$, which means that the model is well-specified.*

The framing (85) implies that optimizing $\mathcal{F}$ can decrease $\mathcal{L}$ and vice-versa. Moreover, in the well specified case, optimizing $\mathcal{F}$ is equivalent to optimizing $\mathcal{L}$. Hence one can use the gradient flow of the MMD defined in (5) to solve (84). One particular setting when (84) is well-specified is the student-teacher problem as in [14]. In this case, a teacher network of the form $\Psi_T(z, \mu)$ produces a deterministic output $y = \Psi_T(z, \mu)$ given an input $z$ while a student network $\Psi_S(z, \nu)$ tries to learn the mapping $z \mapsto \Psi_T(z, \mu)$ by minimizing (84). In practice $\mu$ and $\nu$ are given as empirical distributions on some particles $\Xi = (\xi^1, ..., \xi^M)$ and $X = (x^1, ..., x^N)$ with $\mu = \frac{1}{M} \sum_{j=1}^M \delta_{\xi^j}$ and $\nu = \frac{1}{N} \sum_{i=1}^N \delta_{x^i}$. The particles $(x^i)_{1 \leq i \leq N}$ are then optimized using gradient descent starting from an initial configuration $(x_0^i)_{1 \leq i \leq N}$. This leads to the update equation:

$$x_{n+1}^i = x_n^i - \gamma \mathbb{E}_{z \sim p} \left[ \left( \frac{1}{N} \sum_{j=1}^N \psi(z, x_n^j) - \frac{1}{M} \sum_{j=1}^M \psi(z, \xi^j) \right) \nabla_{x_n^i} \psi(z, x_n^i) \right], \tag{87}$$

where $(x_n^i)_{1 \leq i \leq N}$ are the particles at iteration $n$ with empirical distribution $\nu_n$. Here, the gradient is rescaled by the number of particles $N$. Re-arranging terms and recalling that $k(x, x') = \mathbb{E}_{z \sim p}[\psi(z, x)^T \psi(z, x')]$, equation (87) becomes:

$$x_{n+1}^i = x_n^i - \gamma \nabla f_{\mu, \nu_n}(x_n^i).$$

with $\nabla f_{\mu, \nu_n}(x_n^i) = \left( \frac{1}{N} \sum_{j=1}^N \nabla_2 k(x_n^j, x_n^i) - \frac{1}{M} \sum_{j=1}^M \nabla_2 k(\xi^j, x_n^i) \right)$. The above equation is a discretized version of the gradient flow of the MMD defined in (5). Such discretization is obtained from (21) by setting the noise level $\beta_n$ to 0. Hence, in the limit when $N \to \infty$ and $\gamma \to 0$, one recovers the gradient flow defined in (9). In general the kernel $k$ is intractable and can be approximated using $n_b$ samples $(z_1, ..., z_{n_b})$ from the data distribution: $\hat{k}(x, x') = \frac{1}{n_b} \sum_{b=1}^{n_b} \psi(z_b, x)^T \psi(z_b, x')$. This finally leads to an approximate update:

$$x_{n+1}^i = x_n^i - \gamma \nabla \hat{f}_{\mu, \nu_n}(x_n^i).$$

where $\nabla \hat{f}_{\mu, \nu_n}$ is given by:

$$\nabla \hat{f}_{\mu, \nu_n}(x_n^i) = \frac{1}{n_b} \sum_{b=1}^{n_b} \left( \frac{1}{N} \sum_{j=1}^N \psi(z_b, x_n^j) - \frac{1}{M} \sum_{j=1}^M \psi(z_b, \xi^j) \right) \nabla_{x_n^i} \psi(z_b, x_n^i)).$$

We provide now a proof for Proposition 20:

*Proof of Proposition 20.* Let $\Psi(z, \nu) = \int \psi(z, x) \, d\nu(x)$. By (86), we have: $k(x, x') = \int_z \psi(z, x)^T \psi(z, x') \, ds(z)$ where $s$ denotes the distribution of $z$. It is easy to see that $\mathcal{F}(\nu) = \frac{1}{2} \int \|\Psi(z, \nu) - \Psi(z, \mu)\|^2 \, ds(z)$. Indeed expanding the square in the l.h.s and exchanging the order of integrations w.r.t $p$ and $(\mu \otimes \nu)$ one gets $\mathcal{F}(\nu)$. Now, introducing $\Psi(z, \mu)$ in the expression of $\mathcal{L}(\nu)$, it follows by a simple calculation that:

$$\mathcal{L}(\nu) = \mathcal{L}(\mu) + \mathcal{F}(\nu) + \int \langle \Psi(z, \mu) - m(z), \Psi(z, \nu) - \Psi(z, \mu) \rangle \, dp(z) \tag{88}$$

where $m(z)$ is the conditional mean of $y$, i.e.: $m(z) = \int y \, dp(y|z)$. On the other hand we have that $2\mathcal{L}(\mu) = var(y) + \int \|\Psi(z, \mu) - m(z)\|^2 \, dp(z)$, so that $\int \|\Psi(z, \mu) - m(z)\|^2 \, dp(z) = 2\bar{\mathcal{L}}(\mu)$. Hence, using Cauchy-Schwartz for the last term in (88), one gets the upper-bound:

$$\mathcal{L}(\nu) \leq \mathcal{L}(\mu) + \mathcal{F}(\nu) + 2\bar{\mathcal{L}}(\mu)^{\frac{1}{2}} \mathcal{F}(\nu)^{\frac{1}{2}}.$$

This in turn gives an upper-bound on $\bar{\mathcal{L}}(\nu)$ after subtracting $var(y)/2$ on both sides of the inequality. To get the lower bound on $\bar{\mathcal{L}}$ one needs to use the global optimality condition of $\mu$ for $\mathcal{L}$ from [15, Proposition 3.1]. Indeed, for any $0 < \epsilon \leq 1$ it is easy to see that:

$$\epsilon^{-1}(\mathcal{L}(\mu + \epsilon(\nu - \mu)) - \mathcal{L}(\mu)) = \int \langle \Psi(z,\mu) - m(z), \Psi(z,\nu) - \Psi(z,\mu) \rangle \, \mathrm{d}p(z) + o(\epsilon).$$

Taking the limit $\epsilon \to 0$ and recalling that the l.h.s is always non-negative by optimality of $\mu$, it follows that $\int \langle \Psi(z,\mu) - m(z), \Psi(z,\nu) - \Psi(z,\mu) \rangle \, \mathrm{d}p(z)$ must also be non-negative. Therefore, from (88) one gets that $\mathcal{L}(\nu) \geq \mathcal{L}(\mu) + \mathcal{F}(\nu)$. The final bound is obtained by subtracting $var(y)/2$ again from both sides of the inequality. $\qquad \square$

## G   Numerical Experiments

### G.1   Student-Teacher networks

We consider a student-teacher network setting similar to [14]. More precisely, using the notation from Appendix F, we denote by $\Psi(z,\nu)$ the neural network of the form: $\Psi(z,\nu) = \int \psi(z,x) \, \mathrm{d}\nu(x)$ where $z$ is an input vector in $\mathbb{R}^p$ and $\nu$ is a probability distribution over the parameters $x$. Hence $\Psi$ is an expectation over sub-networks $\psi(z,x)$ with parameters $x$. Here, we choose $\psi$ of the form:

$$\psi(z,x) = G\left(b^1 + W^1 \sigma(W^0 z + b^0)\right). \tag{89}$$

where $x$ is obtained as the concatenation of the parameters $(b^1, W^1, b^0, W^0) \in \mathcal{X}$, $\sigma$ is the ReLU non-linearity while $G$ is a fixed function and is defined later. Note that using $x$ to denote the parameters of a neural network is unusual, however, we prefer to keep a notation which is consistent with the rest of the paper. We will only consider the case when $\nu$ is given by an empirical distribution of $N$ particles $X = (x^1, ... x^N)$ for some $N \in \mathbb{N}$. In that case, we denote by $\nu_X$ such distribution to stress the dependence on the particles $X$, i.e.: $\nu := \nu_X = \frac{1}{N} \sum_{i=1}^{N} \delta_{x^i}$. The teacher network $\Psi_T(z, \nu_\Xi)$ is given by $M$ particles $\Xi = (\xi_1, ..., \xi_M)$ which are fixed during training and are initially drawn according to a normal distribution $\mathcal{N}(0,1)$. Similarly, the student network $\Psi_S(z, \nu_X)$ has $N$ particles $X = (x^1, ..., x^N)$ that are initialized according to a normal distribution $\mathcal{N}(10^{-3}, 1)$. Here we choose $M = 1$ and $N = 1000$. The inputs $z$ are drawn from a uniform distribution $\mathbb{S}$ on the sphere in $\mathbb{R}^p$ as in [14] with $p = 50$. The number of hidden layers $H$ is set to 3 and the output dimension is 1. The parameters of the student networks are trained to minimize the risk in (90) using SGD with mini-batches of size $n_b = 10^2$ and optimal step-size $\gamma$ selected from: $\{10^{-3}, 10^{-2}, 10^{-1}\}$.

$$\min_X \mathbb{E}_{z \sim \mathbb{S}} \left[ (\Psi_T(z, \nu_\Xi) - \Psi_S(z, \nu_X))^2 \right] \tag{90}$$

When $G$ is simply the identity function and no bias is used, one recovers the setting in [15]. In that case the network is partially 1-homogeneous and [15, Theorem 3.5] applies ensuring global optimality. Here, we are interested in the case when global optimality is not guaranteed by the homogeneity structure, hence we choose $G$ to be a gaussian with fixed bandwidth $\sigma = 2$. As shown in Appendix F, performing gradient descent to minimize (90) can be seen as a particle version of the gradient flow of the MMD with a kernel given by $k(x,x') = \mathbb{E}_{z \sim \mathbb{S}}[\psi(z,x)\psi(z,x')]$ and target distribution $\mu$ given by $\mu = \nu_\Xi$. Hence one can use the noise injection algorithm defined in (21) to train the parameters of the student network. Since $k$ is defined through an expectation over the data, it can be approximated using $n_b$ data samples $\{z_1, ..., z_B\}$:

$$\hat{k}(x,x') = \frac{1}{n_b} \sum_{b=1}^{n_b} \psi(z_b, x)\psi(z_b, x'). \tag{91}$$

Such approximation of the kernel leads to a simple expression for the gradient of the unnormalised witness function between $\nu_\Xi$ and $\nu_X$:

$$\nabla \hat{f}_{\nu_\Xi, \nu_X}(x) = \frac{1}{n_b} \sum_{b=1}^{n_b} \left( \frac{1}{M} \sum_{j=1}^{M} \psi(z_b, \xi^j) - \frac{1}{N} \sum_{i=1}^{N} \psi(z_b, x^i) \right) \nabla_x \psi(z_b, x), \qquad \forall x \in \mathcal{X}. \tag{92}$$

Algorithm 2, provides the main steps to train the parameters of the student network using the noisy gradient flow of the MMD proposed in (21). It can be easily implemented using automatic

differentiation packages like `PyTorch`. Indeed, one only needs to compute an auxiliary loss function $\mathcal{F}_{aux}$ instead of the actual MMD loss $\mathcal{F}$ and perform gradient descent using $\mathcal{F}_{aux}$. Such function is given by:

$$\mathcal{F}_{aux} = \frac{1}{n_b} \sum_{i=1}^{N} \sum_{b=1}^{n_b} \left( \texttt{NoGrad}\left(y_S^b\right) - y_T^b \right) \psi(z^b, \widetilde{x}_n^i)$$

To compute $\mathcal{F}_{aux}$, two forward passes on the student network are required. A first forward pass using the current parameter values $X_n = (x_n^1, ..., x_n^N)$ of the student network is used to compute the predictions $y_S^b$ given an input $z^b$. For such forward pass, the gradient w.r.t to the parameters $X_n$ is not used. This is enforced, here, formally by calling the function $\texttt{NoGrad}$. The second forward pass is performed using the noisy parameters $\widetilde{x}_n^i = x_n^i + \beta_n u_n^i$ and requires implementing special layers which can inject noise to the weights. This second forward pass will be used to provide a gradient to update the particles using back-propagation. Indeed, it is easy to see that $\nabla_{x_n^i} \mathcal{F}_{aux}$ gives exactly the gradient $\nabla \hat{f}_{\nu_\Xi, \nu_X}(\widetilde{x}_n^i)$ used in Algorithm 2.

### G.2 Learning gaussians

Figure 2: Gradient flow of the $MMD$ from a gaussian initial distributions $\nu_0 \sim \mathcal{N}(10, 0.5)$ towards a target distribution $\mu \sim \mathcal{N}(0, 1)$ using $N = M = 1000$ samples from $\mu$ and $\nu_0$ and a gaussian kernel with bandwidth $\sigma = 2$. (21) is used without noise $\beta_n = 0$ in red and with noise $\beta_n = 10$ up to $n = 5000$, then $\beta_n = 0$ afterwards in blue. The left figure shows the evolution of the $MMD$ at each iteration. The middle figure shows the initial samples (black for $\mu$), and the right figure shows the final samples after $10^5$ iterations with step-size $\gamma = 0.1$.

Figure 2 illustrates the behavior of the proposed algorithm (21) in a simple setting, and compares it with the gradient flow of the MMD without noise injection. In this setting, the MMD flow fails to converge to the global optimum. Indeed, as shown in Figure 2(right), some of the final samples (in red) obtained using noise-free gradient updates tend to get further away from the target samples (in black). Most of the remaining samples collapse to a unique point at the center near the origin. This can also be seen from Figure 2(left) where the training error fails to decrease below $10^{-3}$. On the other hand, adding noise to the gradient seems to lead to global convergence, as seen visually from the samples. The training error decreases below $10^{-4}$ and oscillates between $10^{-8}$ and $10^{-4}$. The oscillation is due to the step-size, which remained fixed while the noise was set to $0$ starting from iteration 5000. It is worth noting that adding noise to the gradient slows the speed of convergence, as one can see from Figure 2(left). This is expected since the algorithm doesn't follow the path of steepest descent. The noise helps in escaping local optima, however, as illustrated here.

## H   Auxiliary results

**Proposition 21.** *Under Assumption (A), the unnormalised witness function $f_{\mu,\nu}$ between any probability distributions $\mu$ and $\nu$ in $\mathcal{P}_2(\mathcal{X})$ is differentiable and satisfies:*

$$\nabla f_{\mu,\nu}(z) = \int \nabla_1 k(z, x)\, d\mu(x) - \int \nabla_1 k(z, x)\, d\nu(x) \qquad \forall z \in \mathcal{X} \tag{93}$$

*where $z \mapsto \nabla_1 k(x, z)$ denotes the gradient of $z \mapsto k(x, z)$ for a fixed $x \in \mathcal{X}$. Moreover, the map $(z, \mu, \nu) \mapsto f_{\mu,\nu}(z)$ is Lipschitz with:*

$$\|\nabla f_{\mu,\nu}(z) - \nabla f_{\mu',\nu'}(z')\| \le 2L(\|z - z'\| + W_2(\mu, \mu') + W_2(\nu, \nu')) \tag{94}$$

*Finally, each component of $\nabla f_{\mu,\nu}$ belongs to $\mathcal{H}$.*

---
**Algorithm 1** Noisy gradient flow of the MMD
---
1: **Input** $N$, $n_{iter}$, $\beta_0$, $\gamma$
2: **Output** $(x^i_{n_{iter}})_{1 \le i \le N}$
3: *Initialize $N$ particles from initial distribution $\nu_0$* : $x^i_0 \overset{\text{i.i.d}}{\sim} \nu_0$
4: *Initialize the noise level*: $\beta = \beta_0$
5: **for** $n = 0, \ldots, n_{iter}$ **do**
6:     *Sample $M$ points from the target $\mu$*: $\{y^1, ..., y^M\}$.
7:     *Sample $N$ gaussians* : $\{u^1_n, ..., u^N_n\}$
8:     **for** $i = 1, \ldots, N$ **do**
9:         *Compute the noisy values*: $\widetilde{x}^i_n = x^i_n + \beta_n u^i_n$
10:         *Evaluate vector field*: $\nabla f_{\hat{\mu}, \hat{\nu}_n}(\widetilde{x}^i_n) = \frac{1}{N} \sum_{j=1}^{N} \nabla_2 k(x^j_n, \widetilde{x}^i_n) - \frac{1}{M} \sum_{m=1}^{M} \nabla_2 k(y^m, \widetilde{x}^i_n)$
11:         *Update the particles*: $x^i_{n+1} = x^i_n - \gamma \nabla f_{\hat{\mu}, \hat{\nu}_n}(\widetilde{x}^i_n)$
12:     *Update the noise level using an update rule $h$*: $\beta_{n+1} = h(\beta_n, n)$.
---

---
**Algorithm 2** Noisy gradient flow of the MMD for student-teacher learning
---
1: **Input** $N$, $n_{iter}$, $\beta_0$, $\gamma$, $n_b$, $\Xi = (\xi^j)_{1 \le j \le M}$.
2: **Output** $(x^i_{n_{iter}})_{1 \le i \le N}$.
3: *Initialize $N$ particles from initial distribution $\nu_0$* : $x^i_0 \overset{\text{i.i.d}}{\sim} \nu_0$.
4: *Initialize the noise level*: $\beta = \beta_0$.
5: **for** $n = 0, ..., n_{iter}$ **do**
6:     *Sample minibatch of $n_b$ data points*: $\{z^1, ..., z^{n_b}\}$.
7:     **for** $b = 1, ..., n_b$ **do**
8:         *Compute teacher's output*: $y^b_T = \frac{1}{M} \sum_{j=1}^{M} \psi(z^b, \xi^j)$.
9:         *Compute students's output*: $y^b_S = \frac{1}{N} \sum_{i=1}^{N} \psi(z^b, x^i_n)$.
10:     *Sample $N$ gaussians* : $\{u^1_n, ..., u^N_n\}$.
11:     **for** $i = 1, ..., N$ **do**
12:         *Compute noisy particles*: $\widetilde{x}^i_n = x^i_n + \beta_n u^i_n$
13:         *Evaluate vector field*: $\nabla \hat{f}_{\nu_\Xi, \nu_{X_n}}(\widetilde{x}^i_n) = \frac{1}{n_b} \sum_{b=1}^{n_b} (y^b_S - y^b_T) \nabla_{x^i_n} \psi(z^b, \widetilde{x}^i_n)$
14:         *Update particle $i$*: $x^i_{n+1} = x^i_n - \gamma \nabla \hat{f}_{\nu_\Xi, \nu_{X_n}}(\widetilde{x}^i_n)$
15:     *Update the noise level using an update rule $h$*: $\beta_{n+1} = h(\beta_n, n)$.
---

*Proof.* The expression of the unnormalised witness function is given in (1). To establish (93), we simply need to apply the differentiation lemma [32, Theorem 6.28]. By Assumption (A), it follows that $(x, z) \mapsto \nabla_1 k(z, x)$ has at most a linear growth. Hence on any bounded neighborhood of $z$, $x \mapsto \|\nabla_1 k(z, x)\|$ is upper-bounded by an integrable function w.r.t. $\mu$ and $\nu$. Therefore, the differentiation lemma applies and $\nabla f_{\mu, \nu}(z)$ is differentiable with gradient given by (93).

To prove the second statement, we will consider two optimal couplings: $\pi_1$ with marginals $\mu$ and $\mu'$ and $\pi_2$ with marginals $\nu$ and $\nu'$. We use (93) to write:

$$
\begin{aligned}
\|\nabla f_{\mu, \nu}(z) - \nabla f_{\mu', \nu'}(z')\| &= \|\mathbb{E}_{\pi_1}\left[\nabla_1 k(z, x) - \nabla_1 k(z', x')\right] - \mathbb{E}_{\pi_2}\left[\nabla_1 k(z, y) - \nabla_1 k(z', y')\right]\| \\
&\leq \mathbb{E}_{\pi_1}\left[\|\nabla_1 k(z, x) - \nabla_1 k(z', x')\|\right] + \mathbb{E}_{\pi_2}\left[\|\nabla_1 k(z, y) - \nabla_1 k(z', y')\|\right] \\
&\leq L\left(\|z - z'\| + \mathbb{E}_{\pi_1}[\|x - x'\|] + \|z - z'\| + \mathbb{E}_{\pi_2}[\|y - y'\|]\right) \\
&\leq L(2\|z - z'\| + W_2(\mu, \mu') + W_2(\nu, \nu'))
\end{aligned}
$$

The second line is obtained by convexity while the third one uses Assumption (A) and finally the last line relies on $\pi_1$ and $\pi_2$ being optimal. The desired bound is obtained by further upper-bounding the last two terms by twice their amount. $\square$

**Lemma 22.** *Let $U$ be an open set, $q$ a probability distribution in $\mathcal{P}_2(\mathcal{X} \times \mathcal{U})$ and $\psi$ and $\phi$ two measurable maps from $\mathcal{X} \times \mathcal{U}$ to $\mathcal{X}$ which are square-integrable w.r.t $q$. Consider the path $\rho_t$ from $(\psi)_{\#}q$ and $(\psi + \phi)_{\#}q$ given by: $\rho_t = (\psi + t\phi)_{\#}q \quad \forall t \in [0, 1]$. Under Assumption (A), $\mathcal{F}(\rho_t)$ is differentiable in $t$ with*

$$
\dot{\mathcal{F}}(\rho_t) = \int \nabla f_{\mu, \rho_t}(\psi(x, u) + t\phi(x, u))\phi(x, u)\, dq(x, u)
$$

*where $f_{\mu, \rho_t}$ is the unnormalised witness function between $\mu$ and $\rho_t$ as defined in (1). Moreover:*

$$
\left|\dot{\mathcal{F}}(\rho_t) - \dot{\mathcal{F}}(\rho_s)\right| \leq 3L\,|t - s| \int \|\phi(x, u)\|^2\, dq(x, u)
$$

*Proof.* For simplicity, we write $f_t$ instead of $f_{\mu, \rho_t}$ and denote by $s_t(x, u) = \psi(x, u) + t\phi(x, u)$ The function $h : t \mapsto k(s_t(x, u), s_t(x', u')) - k(s_t(x, u), z) - k(s_t(x', u'), z)$ is differentiable for all $(x, u), (x', u')$ in $\mathcal{X} \times \mathcal{U}$ and $z \in \mathcal{X}$. Moreover, by Assumption (A), a simple computation shows that for all $0 \leq t \leq 1$:

$$
\left|\dot{h}\right| \leq L\left[(\|z - \phi(x, u)\| + \|\psi(x, u)\|)\|\phi(x', u')\| + (\|z - \phi(x', u')\| + \|\psi(x', u')\|)\|\phi(x, u)\|\right]
$$

The right hand side of the above inequality is integrable when $z$, $(x, u)$ and $(x', u')$ are independent and such that $z \sim \mu$ and both $(x, u)$ and $(x', u')$ are distributed according to $q$. Therefore, by the differentiation lemma [32, Theorem 6.28] it follows that $\mathcal{F}(\rho_t)$ is differentiable and:

$$
\dot{\mathcal{F}}(\rho_t) = \mathbb{E}\left[(\nabla_1 k(s_t(x, u), s_t(x', u')) - \nabla_1 k(s_t(x, u), z)).\phi(x, u)\right]. \tag{95}
$$

By Proposition 21, we directly get $\dot{\mathcal{F}}(\rho_t) = \int \nabla f_{\mu, \rho_t}(\psi(x, u) + t\phi(x, u))\phi(x, u)\, dq(x, u)$. We shall control now the difference $|\dot{\mathcal{F}}(\rho_t) - \dot{\mathcal{F}}(\rho_{t'})|$ for $0 \leq t, t' \leq 1$. Using Assumption (A) and recalling that $s_t(x, u) - s_{t'}(x, u) = (t - t')\phi(x, u)$ a simple computation shows:

$$
\begin{aligned}
\left|\dot{\mathcal{F}}(\rho_t) - \dot{\mathcal{F}}(\rho_{t'})\right| &\leq L\,|t - t'|\,\mathbb{E}\left[(2\|\phi(x, u)\| + \|\phi(x', u')\|)\|\phi(x, u)\|\right] \\
&\leq L|t - t'|(2\mathbb{E}\left[\|\phi(x, u)\|^2\right] + \mathbb{E}\left[\|\phi(x, u)\|\right]^2) \\
&\leq 3L|t - t'| \int \|\phi(x, u)\|^2\, dq(x, u).
\end{aligned}
$$

which gives the desired upper-bound. $\square$

We denote by $(x, y) \mapsto H_1 k(x, y)$ the Hessian of $x \mapsto k(x, y)$ for all $y \in \mathcal{X}$ and by $(x, y) \mapsto \nabla_1 \nabla_2 k(x, y)$ the upper cross-diagonal block of the hessian of $(x, y) \mapsto k(x, y)$.

**Lemma 23.** *Let $q$ be a probability distribution in $\mathcal{P}_2(\mathcal{X} \times \mathcal{X})$ and $\psi$ and $\phi$ two measurable maps from $\mathcal{X} \times \mathcal{X}$ to $\mathcal{X}$ which are square-integrable w.r.t $q$. Consider the path $\rho_t$ from $(\psi)_{\#}q$ and $(\psi + \phi)_{\#}q$ given by: $\rho_t = (\psi + t\phi)_{\#}q \quad \forall t \in [0,1]$. Under Assumptions (A) and (B), $\mathcal{F}(\rho_t)$ is twice differentiable in $t$ with*

$$\ddot{\mathcal{F}}(\rho_t) = \mathbb{E}\left[\phi(x,y)^T \nabla_1 \nabla_2 k(s_t(x,y), s_t(x',y'))\phi(x',y')\right]$$
$$+ \mathbb{E}\left[\phi(x,y)^T (H_1 k(s_t(x,y), y_t') - H_1 k(s_t(x,y), z))\phi(x,y)\right]$$

*where $(x,y)$ and $(x',y')$ are independent samples from $q$, $z$ is a sample from $\mu$ and $s_t(x,y) = \psi(x,y) + t\phi(x,y)$. Moreover, if Assumption (C) also holds then:*

$$\ddot{\mathcal{F}}(\rho_t) \geq \mathbb{E}\left[\phi(x,y)^T \nabla_1 \nabla_2 k(s_t(x,y), s_t(x',y'))\phi(x',y')\right] - \sqrt{2}\lambda d \mathcal{F}(\rho_t)^{\frac{1}{2}} \mathbb{E}[\|\phi(x,y)\|^2]$$

*where we recall that $\mathcal{X} \subset \mathbb{R}^d$.*

*Proof.* The first part is similar to Lemma 22. In fact we already know by Lemma 22 that $\dot{\mathcal{F}}(\rho_t)$ exists and is given by:

$$\dot{\mathcal{F}}(\rho_t) = \mathbb{E}\left[(\nabla_1 k(s_t(x,y), s_t(x',y')) - \nabla_1 k(s_t(x,y), z)).\phi(x,y)\right]$$

Define now the function $\xi : t \mapsto (\nabla_1 k(s_t(x,y), s_t(x',y')) - \nabla_1 k(s_t(x,y), z)).\phi(x,y)$ which is differentiable for all $(x,y),(x',y')$ in $\mathcal{X} \times \mathcal{X}$ and $z \in \mathcal{X}$ by Assumption (B). Moreover, its time derivative is given by:

$$\dot{\xi} = \phi(x',y')^T \nabla_2 \nabla_1 k(s_t(x,y), s_t(x',y'))\phi(x,y) \tag{96}$$
$$+ \phi(x,y)^T (H_1 k(s_t(x,y), s_t(x',y')) - H_1 k(s_t(x,y), z))\phi(x,y) \tag{97}$$

By Assumption (A) it follows in particular that $\nabla_2 \nabla_1 k$ and $H_1 k$ are bounded hence $|\dot{\xi}|$ is upper-bounded by $(\|\phi(x,y)\| + \|\phi(x',u')\|)\|\phi(x,y)\|$ which is integrable. Therefore, by the differentiation lemma [32, Theorem 6.28] it follows that $\dot{\mathcal{F}}(\rho_t)$ is differentiable and $\ddot{\mathcal{F}}(\rho_t) = \mathbb{E}\left[\dot{\xi}\right]$. We prove now the second statement. Bu the reproducing property, it is easy to see that the last term in the expression of $\dot{\xi}$ can be written as:

$$\langle \phi(x,y)^T H_1 k(s_t(x,y), .)\phi(x,y), k(s_t(x',y'), .) - k(z,.)\rangle_{\mathcal{H}}$$

Now, taking the expectation w.r.t $x'$,$y'$ and $z$ which can be exchanged with the inner-product in $\mathcal{H}$ since $(x',y',z) \mapsto k(s_t(x',y'), .) - k(z,.)$ is Bochner integrable [46, Definition 1, Theorem 6] and recalling that such integral is given by $f_{\mu,\rho_t}$ one gets the following expression:

$$\langle \phi(x,y)^T H_1 k(s_t(x,y), .)\phi(x,y), f_{\mu,\rho_t}\rangle_{\mathcal{H}}$$

Using Cauchy-Schwartz and Assumption (C) it follows that:

$$|\langle \phi(x,y)^T H_1 k(s_t(x,y), .)\phi(x,y), f_{\mu,\rho_t}\rangle_{\mathcal{H}}| \leq \lambda d \|\phi(x,y)\|^2 \|f_{\mu,\rho_t}\|$$

One then concludes using the expression of $\ddot{\mathcal{F}}(\rho_t)$ and recalling that $\mathcal{F}(\rho_t) = \frac{1}{2}\|f_{\mu,\rho_t}\|^2$. $\square$

**Lemma 24.** *Assume that for any geodesic $(\rho_t)_{t \in [0,1]}$ between $\rho_0$ and $\rho_1$ in $\mathcal{P}(\mathcal{X})$ with velocity vectors $(V_t)_{t \in [0,1]}$ the following holds:*

$$\ddot{\mathcal{F}}(\rho_t) \geq \Lambda(\rho_t, V_t)$$

*for some admissible functional $\Lambda$ as defined in Definition 3, then:*

$$\mathcal{F}(\rho_t) \leq (1-t)\mathcal{F}(\rho_0) + t\mathcal{F}(\rho_1) - \int_0^1 \Lambda(\rho_s, V_s)G(s,t)ds$$

*with $G(s,t) = s(1-t)\mathbb{1}\{s \leq t\} + t(1-s)\mathbb{1}\{s \geq t\}$ for $0 \leq s,t \leq 1$.*

*Proof.* This is a direct consequence of the general identity ([58], Proposition 16.2). Indeed, for any continuous function $\phi$ on $[0,1]$ with second derivative $\ddot{\phi}$ that is bounded below in distribution sense the following identity holds:

$$\phi(t) = (1-t)\phi(0) + t\phi(1) - \int_0^1 \ddot{\phi}(s)G(s,t)ds.$$

This holds a fortiori for $\mathcal{F}(\rho_t)$ since $\mathcal{F}$ is smooth. By assumption, we have that $\ddot{\mathcal{F}}(\rho_t) \geq \Lambda(\rho_t, V_t)$, hence, it follows that:

$$\mathcal{F}(\rho_t) \leq (1-t)\mathcal{F}(\rho_0) + t\mathcal{F}(\rho_1) - \int_0^1 \Lambda(\rho_s, V_s)G(s,t)ds.$$

$\square$

**Lemma 25.** *[Mixture convexity] The functional $\mathcal{F}$ is mixture convex: for any probability distributions $\nu_1$ and $\nu_2$ and scalar $1 \leq \lambda \leq 1$:*

$$\mathcal{F}(\lambda\nu_1 + (1-\lambda)\nu_2) \leq \lambda\mathcal{F}(\nu_1) + (1-\lambda)\mathcal{F}(\nu_2)$$

*Proof.* Let $\nu$ and $\nu'$ be two probability distributions and $0 \leq \lambda \leq 1$. Expanding the RKHS norm in $\mathcal{F}$ it follows directly that:

$$\mathcal{F}(\lambda\nu + (1-\lambda)\nu') - \lambda\mathcal{F}(\nu) - (1-\lambda)\mathcal{F}(\nu') = -\frac{1}{2}\lambda(1-\lambda)MMD(\nu,\nu')^2 \leq 0.$$

which concludes the proof. $\square$

**Lemma 26.** *[Discrete Gronwall lemma] Let $a_{n+1} \leq (1+\gamma A)a_n + b$ with $\gamma > 0$, $A > 0$, $b > 0$ and $a_0 = 0$, then:*

$$a_n \leq \frac{b}{\gamma A}(e^{n\gamma A} - 1).$$

*Proof.* Using the recursion, it is easy to see that for any $n > 0$:

$$a_n \leq (1+\gamma A)^n a_0 + b \left( \sum_{i=0}^{n-1} (1+\gamma A)^k \right)$$

One concludes using the identity $\sum_{i=0}^{n-1}(1+\gamma A)^k = \frac{1}{\gamma A}((1+\gamma A)^n - 1)$ and recalling that $(1+\gamma A)^n \leq e^{n\gamma A}$. $\square$