[Reviews · NeurIPS 2019]

Reviewer 1



This paper seems to accomplish two feats at once: it provides a rather deep dive into the specific topic of gradient flows w.r.t. MMD, while it also lays out some original propositions and theorems that establish the paper's main contributions. The first two sections of the paper are excellent and provide a solid introduction to the material the subsequent sections. Per C1, it appears this is fully realized in Proposition 7 in Section 3.2. As an outsider to this level of detail in the field, it is unclear how strigent this assumption is to provide convergence to a global optimum. The proofs seem correct and the other assumptions (e.g. A) are quite mild, but its unclear if this condition can be even checked in a simple case. The authors mention this can be checked in the case of a (nonweighted) negative Sobolev distance, but it would be nice to clarify if such was known for their weighted distance. Per C2, the authors propose a method to avoid local minima both theoretically and empirically. The idea of mollifying the particles before pushing them through grad f is their way of making sure the support of the particles doesn't collapse over time. This is demonstrated empirically in Section 4.2. There are a few questions I have for this experiment: * How sensitive are the choices of beta_n? It appears from a support collapse perspective, you want them large, but if they are too large then they bias the diffusion. * Is it true the validation error eventually plateaus? Is this b/c the beta_n > 0? * Are there any other methods that the noisy gradient method can be compared to, e.g., SGVD? Originality: The paper appears to provide to novel results, including many supporting propositions along the way. It is unclear how readily these results extend to GANs or the training of other NNs. The authors provide a theoretical result in the appendix (Theorem 10) but its empirically not clear how useful these updates are in comparison to standard gradient descent. Quality: The paper appears correct, although I have not read through all the proofs in the appendix (its a 33 page paper!). Clarity: The paper is wonderfully written. Despite the technical nature of the content, it is very clear and approachable for those not completely familiar with the subject matter. Significance: Not being an expert in this field, I have uncertainty in my assessment on the overall impact of this paper. However, the paper is well written and provides two novel ideas relating to gradient flows for MMD, which seem like a worthy contribution to the field. APPENDUM: Upon reading the other reviewers critiques and the author feedback, my score here stands. It isn't exactly clear how much weaker the assumptions used in Proposition 7 are than the state-of-the-art in the literature, so I'm unable to make any meaningful changes.

Reviewer 2



The problem studied by the authors of defining (and proving convergence) of a gradient flow of the MMD with respect to the Wasserstein distance is an important one. As noted in the paper, understanding the behavior of this gradient flow could lead a better understanding of the training dynamics of GANs. Eventually this could lead to better, theoretically sound procedures to train GANS. The convergence analysis employed at a high level follows the steps in Ambrosio, Gigli and Savaré 2008 who studied the gradient flow of KL divergence with respect to Wasserstein distance (studied by JKO initially). While the main ingredients of their analysis such as Proposition 4 and Proposition 5 (via displacement convexity) have direct analogs in the proofs of Ambrosio et al. the presentation here is done well and seems to be technically correct. The authors in Section 4 also present simple extensions to their algorithm by adding noise to regularize and a sample based approximation to the gradients to make their algorithm practically viable. They also provide simple extensions to their theory to motivate these extensions. Overall the paper is well written, and the problem they solve is interesting and important. Some terms however are used before they are defined. For example in Ln. 31 witness functions are undefined. The assumptions used in paper are deep in the appendix in Appendix C without any mention of this in the paper. It would be useful to state these or link to these in the main paper. Further the authors could perhaps some intuition behind the definition and usefulness displacement convexity (also known as geodesic convexity) and how that plays into their proof. ===== Post Author Feedback ========= Thank you for you polite rebuttal. In the discussion, an important point was raised about the locality condition made in in the proof of Proposition 7 and 8. I would urge the authors to motivate and provide an interpretation for this condition (which seems to be hard to check and crucial) in subsequent drafts.

Reviewer 3



For any objective functional of a probability measure, one can write down its Wasserstein gradient flow. When the objective functional is quadratic, the gradient flow is of particular interest and has been studied in the literature: 1. In the interacting particle system in physics, [Carrillo, McCann, Villani, 2006]. 2. In the case of neural networks [Mei, Montanari, and Nguyen, 2018], [Rotskoff and Vandan-Eijnden, 2018]. 3. In the case of KSD flow, [Mroueh, Sercu, and Raj, 2019]. One difficulty of Wasserstein flow of quadratic functional is to establish global convergence result under reasonable assumptions. The new results (which is the main contribution as the authors wrote in the introduction) in this paper are global convergence results in Proposition 7 and Theorem 6. However, these results are quite weak in the sense that it requires unverifiable assumptions on the trajectory. This makes the theory to be like, the gradient flow will converge to global optimal if you start near the global optimal, which is essentially local convergence. Therefore, this “global convergence theory” doesn’t sound satisfactory. In the conclusion section, the authors claimed novelty of introducing the MMD flow. The authors recognized that MMD functional is a quadratic functional of the probability measure of interest, and recognized that existing results (the descending properties, entropy regularization, propagation of Chaos bound of discretizations) on Wasserstein flow of quadratic functional can be applied to MMD functional. Since there are works introducing KSD flow and SVGD relating generative modeling and Wasserstein gradient flow, whether recognizing this relationship is of enough contribution is questionable in reviewer’s point of view. The good side of this paper is, it is well written and properly cited. ---- After reading the response: Thank the authors clarifying the connections and differences of the MMD flow and the results in the literatures. In terms of writings and reference, I think this paper did pretty well. After discussing with the other reviewer, I also believe this paper did a good job on adapting the proof of Wasserstein gradient flow to their settings. This is the reason that I decide to increase my score to 5. My concern is still the trajectory dependent convergence results (Proposition 7, 8 and Theorem 6), which is the new ingredients of this paper. I agree with the author that it is unclear a priori what locality condition is best suited to characterize convergence of the MMD flow. Though, I am not very satisfied with the assumptions in line 218, 219, 268. I am not convinced that, these assumptions and results provide more insights than saying gradient descent will converge to global minimum if it doesn't stuck at bad local min. (Proposition 8 seems providing exponential convergence, but actually the condition \sum \beta_n^2 = \infty with beta_n satisfying Eq. (18) is trajectory dependent and cannot be easily checked. )

[Author Response · NeurIPS 2019]



(a) Training error per second (b) Training error per iteration (c) Test error per epoch (d) Sensitivity to noise (Training) (e) Sensitivity to noise (Test)

Thank you to all the reviewers for their insightful suggestions, which we will implement along with results and
clarifications below. **When do prop. 7 conditions hold:** we relax the condition by restricting the set of test functions
in Eq 15 to functions in the RKHS and exactly obtain the KSD [31], allowing us to compute and verify it in simple
cases. E.g., when both $\mu$ and $\nu_0$ are gaussians and the kernel is $k(x, y) = xy + x^2 y^2$, then $\nu_t$ is also gaussian with
mean $a_t$ and variance $\sigma_t$ satisfying a coupled differential equation. The KSD can be computed in closed form in terms
of $a_t$ and $\sigma_t$. The condition is then verified using standard dynamical systems techniques. A simpler example: linear
kernel; both $\mu$ and $\nu_0$ are gaussians with the same variance. The dynamical system can be solved in closed form and the
condition in Prop. 7 is satisfied. We will add both examples with detailed proofs in the appendix. In the more general
case, weighted negative sobolev distance (WNSD) can be controlled if $\nu_t$ has a density bounded below [35]. Getting
milder conditions remains an open question. **Conditions that depend on the whole trajectory** have been considered
previously: [A, Theorem 6.1], assumes that the first moment remains constant to get a stronger convergence result. We
agree with R.1 that Prop. 7 can be interpreted as a locality condition. Unlike in the finite dimensional case, however, it
is unclear a priori what locality condition is best suited to characterize convergence of the MMD flow. Our contribution
is to use the WNSD to define the notion of locality, which may be useful for further analysis of MMD flow convergence.

**Reviewer 1**. We've added experiments (the figure shows results averaged over 10 runs. Same setting as in Sec G.1)
and now compare with an entropy-regularized flow [30] and KSD flow [31]. (d,e) our noise injection method (red)
is robust to the amount of noise and achieves best performance over a wide region. MMD + diffusion (green) has
worse performance over a much narrower optimal region. KSD (purple) behaves better than MMD (blue) (b and c),
however we (red) still outperform it. Moreover, the computational cost is much higher for KSD (a). We didn't include
SVGD in this comparison since it requires the closed form unnormalised density of the target, contrary to our paper
setting. **Validation error plateaus** even for $\beta_n > 0$, since the kernel is estimated using finitely many random features
in training (empirical version, Eq 42 in Sec B.1). Validation error is estimated using new RFs. **MMD flow vs training**
**NNs** we'll add this clarification: "The MMD flow (without noise) with a finite number of particles is equivalent to
standard GD with a quadratic loss, as shown in Sec B.1 and G.1 and in [37]. Therefore, our results could be used to
analyze convergence of GD in NNs even when a non-linearity is applied after the final activation. This is by contrast
to [37,11] where the final layer must be linear to get convergence results." In the figure, MMD flow (blue) is in fact
equivalent to SGD. Stochasticity is due to the estimation of the kernel using RFs (Eq 42+43 in Sec B.1). Our algorithm
based on noise injection (red) can therefore be applied to train NNs as shown in Alg. 2 of App. G.1. We'll clarify this
in the main paper, along with extensions to more general cost functions.

**Reviewer 2**. We'll gladly implement all suggested clarifications and provide more intuition for displacement convexity.

**Reviewer 3**. As suggested, we will provide more discussion of prior works, and how they differ from ours, as follows.
[Carillo+, Thm. 6.1] provides a convergence results when both potentials satisfy convexity assumptions. This can't be
applied for MMD flow as it requires convexity of either the potential or interaction term. Both terms involve the same
kernel but with opposite signs, so even under convexity of the kernel, a concave term appears and cancels the effect
of the convex term, and [Carillo+, Remark 6.4] fails to hold. Moreover, the requirement that the kernel be positive
semi-definite makes it hard to construct interesting convex kernels. In [30], an entropic regularization is used and allows
to prove convergence to the global optimum. However, the latter is in general different from the global optimum of the
un-regularized loss. To get an accurate solution, small levels of noise are required which can be of limited interest in
practice: green traces, Figure. Our proposed algorithm in (Eq. 21) is different from entropic regularization, and the
global optimum of the MMD remains a fixed point of the algorithm. Qualitatively, the behavior is also very different:
Figure (red). In [37,11], the loss function has a particular structure: '1-homogeneity'.This is well suited for NNs with a
linear final layer and leads to an elegant proof for global optimality. In our case, this corresponds to a kernel $k$ of the
form $k(x, x') = cc' \kappa(\theta, \theta')$ where $x$ and $x'$ are of the form $x = (c, \theta)$ and $x' = (c', \theta')$. However, when a characteristic
kernel is required (to ensure the MMD is a metric), such a structure can't be exploited. KSD flow [31] is also shown to
minimize the MMD. However, those are two different functionals and behave differently: Figure (purple vs blue/red).
See also our App. E.1 discussion on the global optimality condition in [31]. SVGD [28], was introduced as a gradient
flow of the KL w.r.t. a metric [28, Eq. 20] that is not the Wasserstein metric, and requires a closed form target density.
Finally, we emphasize our new noise injected flow in Sec. 4., which improves over the Sec. 3 "vanilla" MMD flow
and has a global convergence result (Prop. 8) that supplants the Prop. 7 conditions. We provide empirical evidence
of its benefits compared to other methods in the figure. The algorithm can be best understood as a generalization of
*continuation methods* to interacting potentials. To our knowledge, our work is the first to propose this approach.

[Meta-Review · NeurIPS 2019]

This paper introduces a variational formulation for Maximum Mean Discrepancy, a generative modeling framework based on RKHS techniques. The formulation is given in terms of a gradient flow in 2-Wasserstein space. While the gradient flow viewpoint is not particularly new in the context of generative modeling, the scope and the quality of the results (convergence toward global optimum; regularization by noise injection; closed-form implementation of the updates) are sufficient for a poster presentation at NeurIPS.